# Is something rotten in the state of Denmark? Cross-national evidence for widespread involvement but not systematic use of questionable research practices across all fields of research

Jesper W. Schneider[1]*, Nick Allum[2], Jens Peter Andersen[1], Michael Bang Petersen[3], Emil B. Madsen[1], Niels Mejlgaard[1], Robert Zachariae[4,5]

1 Danish Centre for Studies in Research and Research Policy, Aarhus University, Aarhus, Denmark, 2 Department of Sociology, University of Essex, Essex, United Kingdom, 3 Department of Political Science, Aarhus University, Aarhus, Denmark, 4 Unit for Psychooncology and Health Psychology (EPoS), Department of Oncology, Aarhus University Hospital, Aarhus, Denmark, 5 Department Psychology and Behavioral Science, Aarhus University, Aarhus, Denmark

* jws@ps.au.dk

**Data Availability Statement:** All relevant data for this study are publicly available from the OSF repository (https://osf.io/rf4bn).

## Abstract

Questionable research practices (QRP) are believed to be widespread, but empirical assessments are generally restricted to a few types of practices. Furthermore, conceptual confusion is rife with use and prevalence of QRPs often being confused as the same quantity. We present the hitherto most comprehensive study examining QRPs across scholarly fields and knowledge production modes. We survey perception, use, prevalence and predictors of QRPs among 3,402 researchers in Denmark and 1,307 in the UK, USA, Croatia and Austria. Results reveal remarkably similar response patterns among Danish and international respondents ($\tau$ = 0.85). Self-reported use indicates whether respondents have used a QRP in recent publications. 9 out of 10 respondents admitted using at least one QRP. Median use is three out of nine QRP items. Self-reported prevalence reflects the frequency of use. On average, prevalence rates were roughly three times lower compared to self-reported use. Findings indicated that the perceived social acceptability of QRPs influenced self-report patterns. Results suggest that most researchers use different types of QRPs within a restricted time period. The prevalence estimates, however, do not suggest outright systematic use of specific QRPs. Perceived pressure was the strongest systemic predictor for prevalence. Conversely, more local attention to research cultures and academic age was negatively related to prevalence. Finally, the personality traits conscientiousness and, to a lesser degree, agreeableness were also inversely associated with self-reported prevalence. Findings suggest that explanations for engagement with QRPs are not only attributable to systemic factors, as hitherto suggested, but a complicated mixture of experience, systemic and individual factors, and motivated reasoning.

**Funding:** JWS 6183-00001B Danish Agency for Science and Higher Education (Ministry of Higher Education and Science) https://ufm.dk/forskning-og-innovation/tilskud-til-forskning-og-innovation/hvem-har-modtaget-tilskud/2016/bevilling-til-forskning-i-dansk-forskningsintegritet-fra-styrelsen-for-forskning-og-innovation The funders had no role in study design, data collection and analysis, decision to publish, or preparation of the manuscript."

**Competing interests:** NO authors have competing interests.

## Introduction

The scientific literature in several fields is seemingly swamped with false-positive claims [1], and a so-called reproducibility crisis has been declared [2]. The widespread use of questionable research practices (QRPs) is believed to be the main culprit for this untenable situation [3]. That is at least the impression one gets from popular media and rapidly growing scholarly literature reporting seemingly serious challenges to the scientific enterprise [e.g., 4, 5]. In this view, QRPs, their prevalence, and their consequences are confined to a select few manipulative, undisclosed practices which are exploited to seek 'statistically significant' and thus publishable results [6]. Nevertheless, the concept of QRP is far more comprehensive. It covers a wide range of diverse practices with potentially equally serious consequences for the trustworthiness and integrity of the scientific enterprise and its outcomes [7]. Therefore, there is a need for a more comprehensive perspective if we are to understand QRPs better, as well as their roots, use, prevalence, and impacts.

When, in 2005, Martinson and colleagues warned that, surprisingly, many US-based scientists funded by the *National Institute of Health* (NIH) 'misbehaved' [8], it marked an emerging reorientation in the discussions and studies on breaches of research integrity. They called for the need to change focus from spectacular but presumably rare cases of misconduct to a broader range of more "mundane 'regular' misbehaviours" viewed as questionable research practices [8], defined as actions that deviate from traditional values and standards of the research enterprise [9, 10]. They are not acts of misconduct, commonly defined as fabrication, falsification, and plagiarism (FFP) [10]. Instead, they are actions that violate the fundamental tenets of research, performed inadvertently or intentionally, occurring at every stage in the research process, and falling on a spectrum of severity from sloppy to detrimental [11]. QRPs are 'ethical shades of grey' between acceptable and unethical practices that offer considerable latitude for rationalisation and self-deception [12]. Individual QRPs often resist simple definitions and may even be defensible in specific contexts [13, 14]. They may include acts such as claiming authorship when not warranted, flawed peer reviewing, selective citing and deceptive reporting, overselling results, and various manipulative statistics popularly referred to as *p*-hacking [6, 7]. QRPs produce bias in everything from peer reviewing [e.g., 15] and authorship credits [e.g., 16–18] over citation networks and indicators [e.g., 19–23] to published research findings [e.g., 1, 24]. QRPs are presumed to be much more prevalent than individual acts of misconduct and are, therefore, seen as more damaging to research integrity [11]. If this is the case, the validity of knowledge claims in the literature will be undermined [7, 25], resources will be wasted [26], unnecessary harm may be inflicted, scientific norms will be challenged [27], trust between researchers will be eroded, and unfair reward structures evolve [28, 29].

Most researchers appear to agree that widespread breaches of research integrity are detrimental to the credibility of the scientific enterprise and likewise assume that an academic climate of 'perverse incentives' and 'hyper-competition', i.e. corrupting systemic structures, are the primary explanations for such breaches [e.g., 8, 25, 29–31]. While such claims seem 'too good to be false' [32], they are also very convenient as they remove the focus from one's responsibility for potentially questionable behaviours [33].

To varying degrees, systemic challenges are undoubtedly real and stimulate motivated reasoning among scientists [34], but the question remains whether they are sufficient for explaining individual differences in QRPs. It does not follow logically that the susceptibility to negative impacts on behaviours from such challenges should be uniformly distributed across individuals. Individual-level engagement with QRPs may also be associated with individual factors. In particular, to the extent that the engagement with certain QRPs reflects a personal choice, it may be influenced by the features known to promote dishonest behaviour in other

domains of life [35]. A first step, therefore, would be to correlate self-reported prevalence rates with personality traits. While research into misconduct has focused on the deviant behaviour of 'bad apples' [36] with assumed sinister personalities [37] or narcissistic traits [38], the presumed widespread nature of QRPs suggests that it may be more relevant to assess the role of personality variation within the normal range. The Big Five Taxonomy is the most widely used approach to personality variation [39], which identifies five general personality trait dimensions. Low levels on two dimensions, conscientiousness and agreeableness, have been consistently associated with various types of academic dishonesty [40–42].

Nevertheless, popular perceptions of QRPs still rest upon restricted and often fragmentary evidence combined with strong presumptions. Before we can assess their full impact on the scientific enterprise, we must explore the various types of QRPs, their perceptions of them across fields and different research settings, their prevalence, and their interplay and potential associations with systemic and individual factors. To that end, we present novel findings from the hitherto most comprehensive systematic study of QRPs. We aim to present a framework that enables broader reflections on QRPs, how we examine them, and linking them to more general discussions of challenges to the scientific enterprise, and not just, as seem predominant, issues of reproducibility, although these are certainly also important.

## A review of the literature on survey studies of QRPs

While surveys remain the most feasible approach to large-scale, comprehensive assessments of the perception, use, and prevalence of QRPs, asking people to answer sensitive questions about their behaviours involves several threats to validity. The available studies demonstrate that examining the perception of QRPs, let alone estimating their prevalence, is challenging and has led to contested findings [43]. In this section, we review the key methods for measuring the use and prevalence of QRPs, attempts to address issues of measurement validity, including social desirability, and, finally, findings on the potential causes of self-reported use of QRPs.

## Studies of the spread of QRPs

A fundamental assumption behind the dominant view of QRPs is that they are widespread. Martinson and colleagues [8] reported that 33% of their respondents admitted to having engaged in at least one of ten QRP-related behaviours within the last three years, compared to slightly under 2% who admitted outright misconduct. The extent of QRPs worried the authors, and they even suggested that the prevalence could be underestimated due to the nature of self-reporting. Several survey-based studies of QRPs have targeted specific populations of medical, social or behavioural researchers. The studies can be divided into two groups. The first is a heterogeneous group of studies using a wide range of items and scales examining misconduct and various types of QRPs [e.g., 8, 12, 44–59]. The findings from these studies are often difficult to compare, but authors generally express their concern about surprisingly many of their respondents who acknowledge questionable or unacceptable behaviours. Despite seemingly incommensurate results, an early meta-analytic attempt claimed prevalence at 34% [60], and recently, a similar meta-study claimed a pooled prevalence estimate of 12.5% [61]. In both studies, prevalence refers to respondents who admit to using at least one QRP.

The second group of studies include more recent and comparable surveys that employ a more homogenous but also a more restricted set of QRP items examining undisclosed manipulative practices used for chasing statistically significant results [13, 62–70]. The earliest and most cited of these studies [65] selected what has since become a canonical set of nine QRPs and surveyed US psychologists, asking them to self-report whether they had ever engaged in each of these practices. Ninety-four per cent admitted that they had used at least one of these

QRPs, and the mean self-reported rate across all QRP items was 42%, with rates for the individual QRPs ranging from 17% to 57%. Based on these findings, the authors [65] claimed evidence for the widespread use of QRPs and suggested that some practices might even constitute the prevailing research norm. Together with [6], this report [65] has shaped the current dominating perception of QRPs as consisting mainly of manipulative statistical practices. Importantly, the study has also become the often cited evidence for the popular belief of widespread use of QRPs in general and their influential role in the so-called reproducibility crisis in particular [5]. Subsequent studies have adopted the same approach as in [65] with few adjustments and find self-reported rates between 22% and 42%, and rates for the comparable individual QRP items varying between 9 and 64% [e.g., 13, 62–64, 66–68, 70, 71].

### Measurement challenges

Several challenges have been raised against such findings. Most pronounced are differences in the recall period for when the practice may have occurred [60, 63]. Furthermore, there seem to be considerable conceptual ambiguities which influence the discussion of prevalence. Fiedler and Schwarz [63] argue that prevalence cannot be inferred from the proportion of respondents who admit using such practices at least once during their academic career. Instead, use and prevalence are two distinct quantities with different implications. Fiedler and Schwarz contend [63] that a proper estimate of QRP prevalence needs to multiply the proportion of respondents admitting use by the self-reported rate of repeated practice over time. They argue that such an approach produces considerably lower estimates and that prevalence, when conflated with 'having done this at least once', perpetuates inflated and overestimated prevalence rates, leading to the highly pessimistic claims about widespread and likely normative use of QRPs [64]. Despite this critique of the dominating approach, only one recent study seems to have used this amended conception of prevalence [71]. A few studies have reported repetition rates for those who admit use based on categorical Likert scales [13], self-reported percentage estimates [70], or, most recently, prevalence based on the percentage of QRPs having a Likert score at the highest end of the 7-point scale [58].

The primary outcome reported in nearly all surveys is self-reported use, where respondents were asked whether they had ever used a particular QRP. It is also evident that these surveys do not differentiate between use and prevalence despite their alleged conceptual difference [63]. As this makes clear, it is crucial to transparently describe *which* prevalence is addressed in a specific study [63]. In the present study, self-reported use indicates *whether* respondents have used a practice in recent publications. Self-reported use, therefore, provides information on respondents' involvement in QRPs. Self-reported prevalence is complimentary; in our case, it indicates the frequency of use in recent publications by those who admit to using it.

Self-reporting is generally presumed to be underreported due to social desirability. However, results from studies using 'truth-telling' instruments [65] to counter such underreporting do not seem to deviate much from normal responses, and most studies do not employ them. Surveys also often ask respondents to guesstimate the prevalence of QRPs among colleagues in their scientific communities. Guesstimates, or perceived prevalence, are presumed to provide interesting insights into perceptions of QRPs, such as beliefs regarding the social acceptability of research practices [e.g., 13, 68–70]. Fraser and colleagues [13] suggest that when perceived prevalence is substantially higher than self-reported prevalence, it may indicate that such QRPs are seen as less socially acceptable. Bakker and colleagues [69] claim that self-reported use has a strong positive influence on perceived prevalence, arguing that if researchers believe that QRPs are widespread, they are also less likely to think that they are doing anything inappropriate.

## Potential predictors of QRPs

Few studies have explored associations between patterns of self-reported QRPs and potential explanatory factors that may motivate such behaviours [e.g., 50, 51, 58, 72, 73]. Current findings suggest that publication pressure, a perceived systemic challenge, positively correlates with self-reported use of QRPs [e.g., 58, 73]. Associations with other factors, such as work conditions, gender and career stage, are less clear. Only two recent studies have examined individual factors. Here, aspects of personal motivation such as achievement goals [72] or a 'strong sense of justification' [71] seem to predict involvement in QRPs.

Linking involvement in irresponsible research behaviours empirically to potential underlying explanatory factors is challenging, and exploring correlates is a crucial first step in identifying or excluding potential underlying factors affecting QRPs. Furthermore, examining the possible interplay between systemic and individual factors seems relevant. In the current research literature on QRPs, systemic and individual factors are most often treated separately, if at all [74].

Taken together, as this review indicates, while widespread use of QRPs may be harmful to the scientific enterprise, also beyond issues of reproducibility, our current knowledge of QRPs remains fragmentary. This is due to the relatively narrow set of QRPs and research areas examined, the ambiguity around measuring use and prevalence and the incomplete understanding of associations between systemic and individual factors and self-reported QRPs. These limitations make it difficult to assess to what extent researchers, irrespective of knowledge production modes, fields, and national settings, are involved in QRPs broadly and to what extent the use of particular QRPs is systematic and seemingly normative. To mitigate some of these challenges, we present findings from a systematic study of QRPs based on a nationwide survey of Danish university-based researchers and cross-validated with responses from an identical survey among researchers in four additional countries. We present the first set of evidence comparing perceptions and self-reporting of diverse practices across main scientific fields. We also report findings on the relative role of systemic and individual factors associated with the self-reported prevalence of such practices.

## Methods

Our survey targeted all researchers from the postdoc level and upwards across all research fields at the eight Danish universities and a comparable set of 12 universities across four other countries. International universities were purposively selected from Austria, the United Kingdom, Croatia, and the US, representing different research systems in scale and structure. The International universities are kept anonymous. The results presented here focus on the perception, use, prevalence, and predictors of QRPs and are part of a larger research project: 'Practices, Perceptions, and Patterns of Research Integrity' (PRINT). Further information about the project, the complete questionnaire, supplementary materials, data, and ethical approval are available at OSF (https://osf.io/rf4bn/). The questionnaire is comprehensive and can support multiple studies, of which this is one. While our project is inherently descriptive and exploratory, we initially chose to preregister to reflect our commitment to transparency (https://osf.io/75g3d). However, in hindsight, our broad preregistration does not align particularly well with the narrow focus of this exploratory study. Developments and circumstances, therefore, have led to six deviations from what was initially outlined. First, we do not report outcomes according to the primary research fields. As one of our reviewers pointed out, aggregated analyses are hardly valid if respondents are not exposed to potentially the same set of QRP statements. This issue arises when grouping by main research fields, leading us to remove these results. Second, for clarity, we have reduced the number of QRP categories from twelve to

eleven. Third, the scope of this study limits us to addressing only two of the four preregistered outcome variables: self-reported use and perceived prevalence. Fourth, we discarded the list experiment results because the implementation was deficient in several respects, which we cannot adequately explain. The decision is supported by similar challenges faced in other surveys on integrity [99]. Fifth, we changed the reported regression model to one not preregistered to ease interpretation, noting that results under the preregistered model are similar. Sixth, we discarded four items from the questionnaire initially intended to capture perceptions of peer review and prestigious publication outlets, along with their potential link to QRPs, to explore them as a predictive systemic factor. Unfortunately, these items and their formulations were problematic for this purpose, resulting in conceptual ambiguity and misalignment with the two other systemic factors included (see survey instrument section below for further elaboration).

## Survey participants

We surveyed all major scholarly fields and knowledge production modes. We harvested publicly available names, positions, department affiliations, and email addresses of potentially eligible researchers at the 20 included research institutions in the spring of 2018. PhD students were omitted as we wanted to ensure that respondents had some recent publication experiences. Data were manually screened to remove apparent non-eligible persons. Sampling commenced on October 10, 2018. Invitations with informed consent were sent out to 15,464 researchers at the Danish universities and 32,450 researchers at the international universities. Reminders were administered four times to the Danish and three times to the international researchers at an interval of approximately 14 days, varying the actual weekday and time of day. Data collection was terminated in early December 2018. A total of 3,402 researchers at the Danish universities completed the questionnaire (response rate = 22%). An additional 747 researchers in the Danish sample engaged with but did not complete the questionnaire. A total of 1,307 researchers at the international universities completed the questionnaire (response rate = 4%) (Austria 3.5%, UK 5%, Croatia 4.2%, and US 2%), with an additional 310 researchers engaging with but not completing the questionnaire. Only respondents with completed questionnaires were included in the analyses, and all were anonymised when sampling was completed. Of the 3,402 respondents who completed the Danish survey, 33% identified as female, 63% as male, 1% identified as non-binary, and 3% preferred not to respond. The corresponding distribution of the international respondents was 39% female, 56% male, 1% non-binary and 4% preferring not to respond. The response rates of the two sets are markedly different. The response rate for the Danish set most likely benefits from public awareness of the survey. The difference in response rates was expected, and our intention with the international survey was primarily to compare and examine response patterns in the Danish survey. While nonresponse bias is rarely significantly related to the nonresponse rate [75], given our focus on dubious or irresponsible research behaviours, nonresponse is most likely to have resulted in some bias.

## Survey instrument

Central to the survey is a pool of 25 potential QRP statements pragmatically assigned to at least one of 11 overall categories (Table 1, see S1 Table for full wording of QRP statements in the questionnaire). As we aimed to examine QRPs across all research fields, we selected a broad pool covering different aspects of the research process and knowledge production modes in different settings. The selection of practices and formulation of statements were informed by reviewing the literature, and suggestions from 22 focus-group interviews carried out in the

**Table 1. Potential QRRs and pragmatic categories used in the survey.**

| No. | Categories | Potential QRPs |
|---|---|---|
| 1 | Authorship | Honorary authorships |
| 2 | Authorship | Fail to offer deserved authorship to collaborators |
| 3 | Transparency | Not disclosing relevant conflicts of interests |
| 4 | Selective analysis | Collect more data if results are non-significant |
| 5 | Selective analysis | Undisclosed data dredging, p-hacking |
| 6 | Recycling | Deliberately publishing redundant work |
| 7 | Citing practices | Cite literature without reading it |
| 8 | Misleading reporting | Claim to have used a qualitative approach appropriately when this was not the case |
| 9 | Transparency | Avoid sharing data, code, protocol, etc. requested by colleagues |
| 10 | Reviewing | Agree to review a manuscript knowing that you have inadequate expertise |
| 11 | Reviewing | Lack of sufficient effort when reviewing |
| 12 | Reviewing | Submitted a biased review report that evaluated the manuscript unfairly |
| 13 | Recycling | Reuse previously published data without disclosure |
| 14 | Recycling | Salami-slicing publications |
| 15 | Citing practices | Cite irrelevant literature to please |
| 16 | Citing practices | Selective over-citing of own publications |
| 17 | Citing practices | Disregard citing relevant contradictory works |
| 18 | Selective reporting | Cherry-pick what supports a hypothesis and disregard that which does not |
| 19 | Selective reporting | Deliberately refrain from reporting findings that could weaken or contradict own theories, hypotheses, or findings. |
| 20 | Spin | Overselling results |
| 21 | Selective analysis | HARKing in confirmatory quantitative studies |
| 22 | Selective analysis | HARKing in confirmatory qualitative studies |
| 23 | Misleading interpretation | Do not distinguish between statistical and practical significance |
| 24 | Misleading interpretation | Report non-significant findings as evidence for no effect |
| 25 | Plagiarism | Plagiarising other researchers' unpublished ideas |

Note the categorisation is pragmatic. Several QRPs could be placed in other or multiple categories.

autumn of 2017 in Denmark [76]. All statements were selected because some had previously identified them as questionable, whether done intentionally or not. We strove to cover a broad spectrum of practices across various phases in the research process, from planning a study, analyses and reporting to collaboration and reviewing.

The 25 chosen practices represent a diverse set of research activities that can be pragmatically grouped into issues of plagiarism, biased reviewing, unfair authorship, lack of transparency, recycling, biased citing practices, selective analyses (e.g., p-hacking, HARKing), selective reporting, spin, and misleading reporting and misleading interpretation (e.g., in relation to 'statistical significance').

Respondents were first asked to choose their preferred research approach among four options to target presumably relevant QRP statements. This selection process is illustrated in Fig 1. Table 2 shows the distribution of respondents according to their preferred research approach.

The 25 QRPs were a priori linked to these four research approaches according to presumed relevance: 11 QRPs were deemed relevant for all research approaches (underlined numbers in Fig 1), and the remaining 14 were relevant to one or more approaches. The QRPs were then allocated based on rules and random assignment (S2 and S3 Tables in S1 File). Each respondent was presented successively with nine QRP statements, some mandatory and others

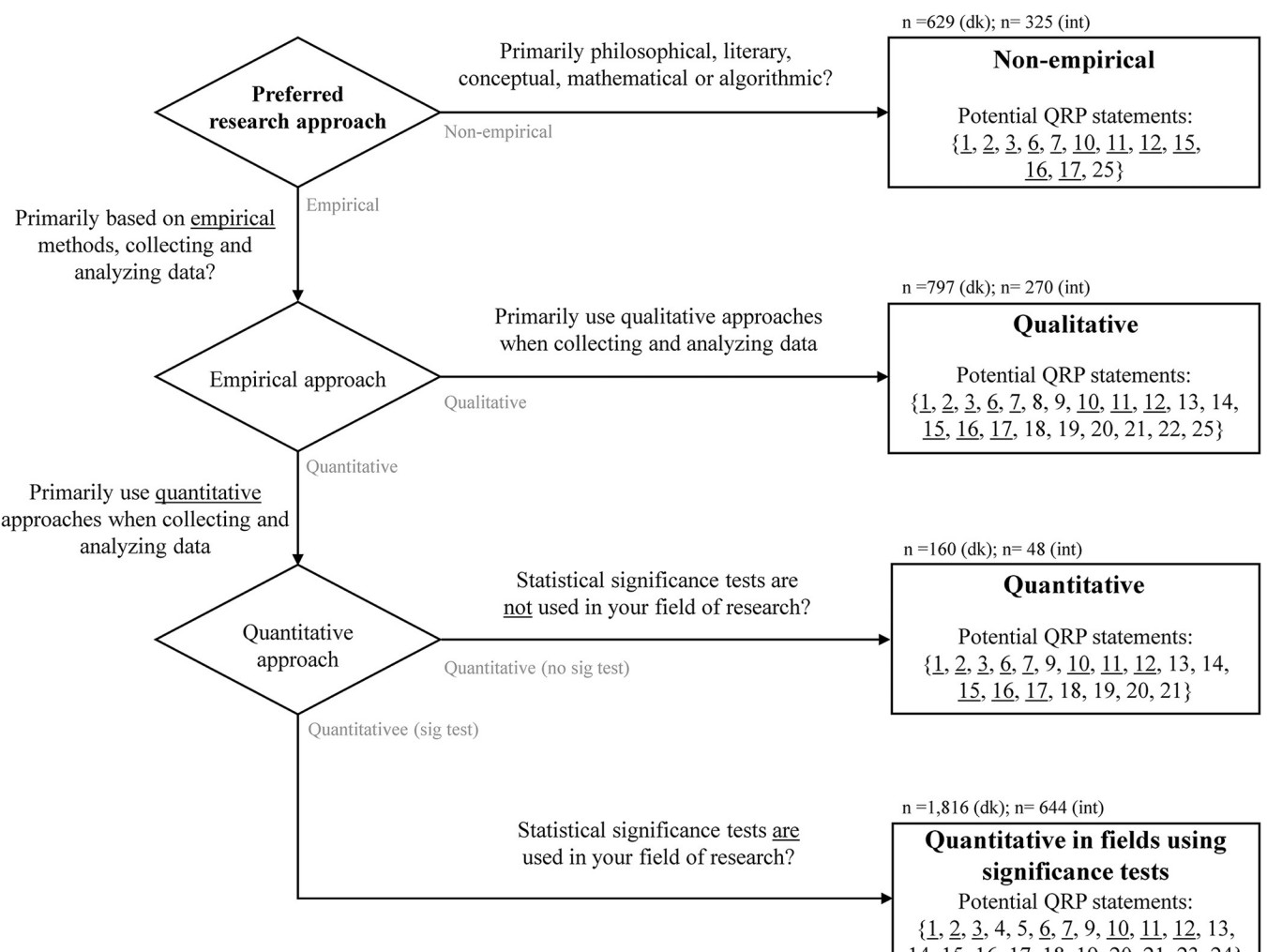

**Fig 1. Choice of preferred research approach among four options to target potentially relevant QRP statements.** The respondents had four options: non-empirical, qualitative, quantitative, and quantitative in fields using significance tests. The numbers of associated potential QRP statements and the sample sizes (n) in the two surveys are shown for each research approach. Underlining the QRP number means that it is common to all. *Choice process*: Initially, respondents were asked to choose between non-empirical or empirical approaches. Those who selected empirical were then asked if their primary empirical approach was qualitative or quantitative. Finally, respondents who identified quantitative as their approach were further inquired whether statistical significance tests are utilised in their field of research. This placed respondents into one of four main categories of preferred research approaches. Respondents in 'quantitative in fields using significance tests' were allowed to discard a QRP statement related to 'significance testing' if they did not use such tests in their recent publications.

**Table 2. Distribution of respondents according to preferred research approach.**

| Respondents | Non-empirical | Empirical, quantitative | Empirical, quantitative using statistical significance testing | Empirical, qualitative | Total |
|---|---|---|---|---|---|
| Danish | 629 (18%) | 160 (5%) | 1,816 (53%) | 797 (23%) | 3,402 |
| International | 325 (25%) | 48 (4%) | 664 (51%) | 270 (21%) | 1,307 |
| Total | 954 | 208 | 2,480 | 1,067 | 4,709 |

In parenthesis are percentages of the total sampling.

**Table 3. Questionnaire item formats for perceived prevalence in the recent literature and self-reported use and prevalence in recent publications.**

| 1. Perceived prevalence | "*To what extent do you believe this practice is used in the recent publications in your specific field of research?*"<br>• The respondents rated this belief on a seven-point scale with two anchors, from "in no recent publications" to "in all recent publications".<br>• Respondents were allowed to mark "unable to answer". |
|---|---|
| 2. Self-reported use | "*To what extent is this practice used in your recent sole or co-authored publications?*"<br>• The respondents rated this belief on a seven-point scale with two anchors, from "in no recent publications" to "in all recent publications".<br>• Respondents were allowed to mark "unable to answer". |

randomly assigned from the eligible pool given the chosen research approach. Respondents were asked to estimate their perception of prevalence in the recent literature they engaged with and subsequently self-reported use and prevalence in their recent publications. They were asked the two questions in the same order for each of the nine QRP statements allocated to the respondent (Table 3).

QRP items were framed somewhat differently from previous studies by focusing on publications rather than persons and by narrowing the recall period for the occurrence of a QRP to 'recent publications'.

Four batteries were included in the questionnaire to examine possible systemic and individual predictors of self-reported prevalence. We initially envisioned three systemic/institutional factors that could predict engagement with QRPs in our subsequent modelling. One factor that captured the respondents' degree of perceived pressure; one that captured the perception of local research cultures, especially concerning a focus on quality assurance and integrity; and finally, a factor that captured the perception of peer review and publishing in prestigious outlets, and how this could affect research integrity. These presumed latent constructs were represented by 16 statements in the questionnaire, grouped into three: 'Perceived pressure' (6 items), 'Peer review and publishing' conditions (4 items) and perceived 'local research culture' (6 items) (items are presented in S4 Table in S2 File).

All three topics were measured with seven-point Likert scales with strongly agree and strongly disagree as anchors and a neutral mid-point (S5 Fig in S2 File). We expected that these groups of items would form three latent constructs to be used as predictors of engagement with QRPs in our subsequent modelling. However, the 'peer review' factor was eventually discarded. Upon deeper analysis, we realised the items were not well-suited for the intended purpose. As designed, the 'peer review' factor does not align as directly with the perceived mechanisms influencing QRP engagement as 'perceived pressure' and 'local research culture' do. It presents an indirect and complex relationship with QRPs compared to the direct, clear connections of the other two. These directly relate to conditions that might compel researchers to compromise integrity. Both 'perceived pressure' and 'local research culture' have high predictive relevance since external and internal pressures are often cited as drivers of QRPs, making 'perceived pressure' likely to be a significant predictor of their use. Conversely, a supportive research culture is logically associated with lower incidences of QRPs, making this factor a plausible protective factor against them. While peer review and intense publication competition may compromise integrity, the way the items were formulated, capturing broad sentiments rather than specific proclivities for, or behavioural inclinations towards, QRP engagement, ultimately forfeited the opportunity to reflect that accurately. Thus, including it would compromise the model's interpretive clarity and predictive relevance. The reliability of the remaining two factors was examined using Cronbach's α, 'perceived pressure' (0.69 (DK), 0.73 (INT)) and 'local research culture' (0.80 (DK), 0.85 (INT)).

To examine individual-level factors' potential association with QRP involvement, we used the Big Five Taxonomy [77]. To limit the time needed to complete the survey to about 15 to 20 minutes, we applied the TIPI 10-item inventory [78]. Although the psychometric properties are somewhat inferior to the standard multi-item instruments, TIPI is deemed adequate for situations where brief measures are needed [78]. Finally, the respondents were asked two demographic questions: gender and year of PhD degree. S6-S9 Figs in S2 File show correlations between predictors and predictors and QRPs for the Danish and international surveys.

## Measures

The Danish and international surveys were analysed separately. The use of QRPs was assessed on two 7-point numerical rating scales assumed to be approximately linear. Respondents were first asked to guess and grade the intensity of use of a QRP in recent publications in their specific field of research from 'in no recent publications' (0) to 'in all recent publications' (6). They were then asked to self-report the intensity of using the same QRP on a similar scale with identical anchors. From this, we derived three outcome measures: Perceived prevalence in the field, self-reported use, and self-reported prevalence. Self-reported use was the proportion of non-zero responses, i.e. respondents admitting having used a QRP at least once in recent publications. Self-reported prevalence reflects the frequency of use in recent publications among respondents admitting use at least once. Self-reported prevalence was obtained by multiplying the proportion of non-zero responses with the mean repetition rate by which the respondents self-reported the practice on a 7-point scale. To convert the scale to proportions, we monotonously transformed the rating scale (0–6) using the POMP method, where scores are represented as the proportion of the maximum possible [79]. For example, for a certain QRP, 20% of respondents admitted use. Their mean repetition rate is calculated from the non-zero integers on the rating scale (1–6); let us assume that the mean is 1.9. We transform 1.9 to a percentage using POMP $\left(\frac{(1.9-0)}{(6-0)}\right) \times 100 = 32.3\%$. Finally, we multiply the proportion of non-zero responses (20%) with the mean repetition rate (32.3%), and the self-reported prevalence for the QRP is then 6.5%. The perceived prevalence measure was derived in a similar way.

## Analyses

We present two primary analyses. First, we conduct descriptive analyses to examine the distribution of responses to the 25 QRP statements from the two surveys, focusing on self-reported use, self-reported prevalence, and perceived prevalence (Figs 2–4). Subsequently, exploratory analyses were carried out to examine potential predictors of self-reported prevalence among the respondents. These analyses were performed on five subsets of respondents according to their preferred research approach. Since respondents were subjected to different subsets of the 25 QRPs depending on their preferred research approach (see Fig 1), aggregate analyses based on all QRPs combined are not considered valid. Instead, analyses were conducted separately for the four preferred research approaches. Within these approaches, all respondents were, in principle, exposed to the same set of QRP statements; some statements were mandatory for the specific research approach, while others were randomly assigned from a shared pool of 11 QRPs available to all respondents regardless of their preferred research approach. The latter also enables a fifth subset for analysis, namely all respondents in the surveys, albeit on the restricted pool of 11 QRPs to which they all, in principle, have been randomly exposed.

Given that randomisation is effective, the effects of random assignment and missing items will cancel out at the aggregate level of each subset examined. This allows for a valid comparison of response patterns within each of the five subsets. By analysing responses within each

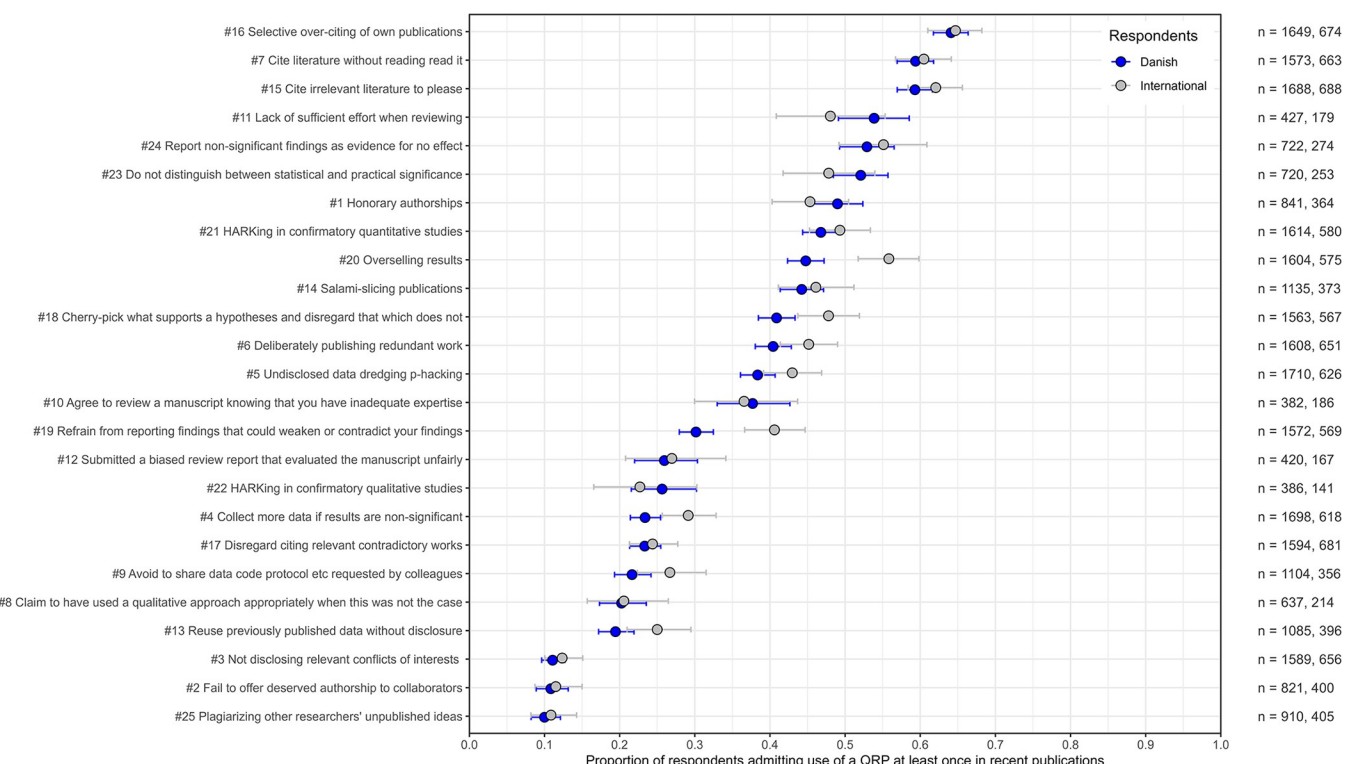

**Fig 2. Self-reported use of 25 potential QRPs from the Danish and international surveys.** QRPs are ranked according to frequency in the Danish survey. Proportions are reported with 95% Bayesian highest probability densities. The number of respondents (n) for each QRP are shown to the right, Danish and international, respectively.

subset, we inherently controlled for the variation in QRP exposure. This makes it easier to interpret differences or similarities in response patterns as more directly related to the respondents' perceptions and behaviours rather than the variability in which QRPs they were asked about.

We explored the associations of systemic, individual, and demographic factors with self-reported prevalence of QRPs using a Bayesian implementation of the Linear Probability Model (LPM) [80]. The outcome variable is the overall self-reported prevalence with QRPs for the individual respondent measured as a proportion in the interval 0 to 1. LPM uses ordinary least squares on a bounded outcome variable, so we compared findings from the LPM models with fractional regressions [81] (S10-S17 Tables in S2 File). As the results were similar, we report probability changes in the predictors from the LPM models as they are more intuitive to interpret than log odds from fractional regressions. The predictors included two systemic factors ('perceived pressure' and 'local research culture'), the five personality factors Openness, Conscientiousness, Extraversion, Agreeableness and Neuroticism (OCEAN), as well as two demographic factors (gender and academic age, i.e., time from PhD to survey year). All predictors were carefully chosen. As already indicated, the systemic factors included have high predictive relevance since pressures are often cited as a cause for breaches in integrity [e.g., 29], and a supportive research culture is frequently pointed to as a plausible protective [e.g., 58]. Likewise, gender and academic age have been examined before, although their predictive value is unclear [e.g., 73]. What is novel in our approach is to combine these relevant systemic and demographic predictors with personality factors in the normal range. We conjecture that propensities for engagements in QRPs are a complicated mixture of experience, systemic and individual factors.

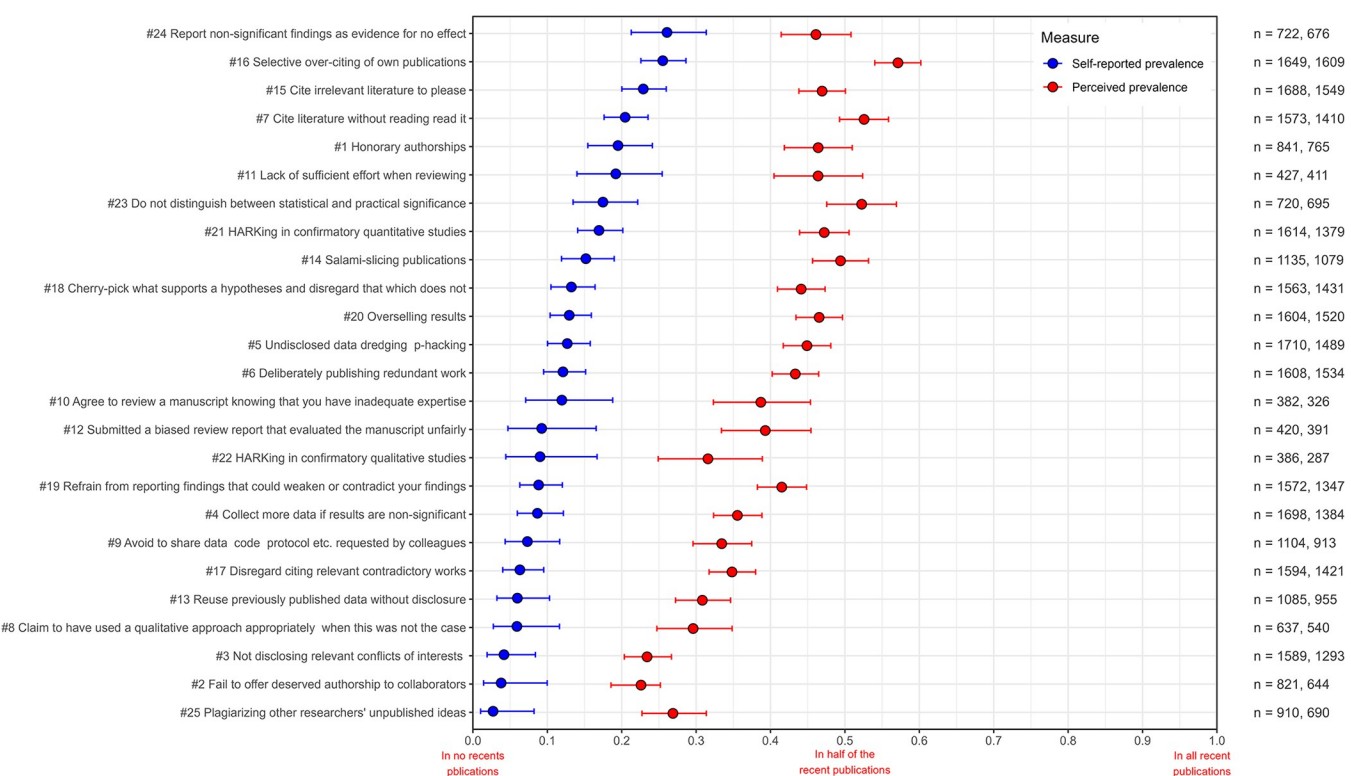

**Fig 3. Estimates of the self-reported (blue) and perceived (red) prevalence of QRPs in recent publications from the Danish survey.** QRPs are ranked according to self-reported prevalence. Proportions are reported with 95% Bayesian highest probability densities. The numbers of respondents (n) for each QRP are shown to the right, self-reported and perceived prevalence, respectively.

All continuous predictors are mean-centred and standardised. The Bayesian models were implemented in *R* [82] with the *brms* package (v 2.16.1) [83] using weakly informative priors, $\mathcal{N}(0, 1)$. This prior is balanced. It does dampen unreasonable parameter values but is not so strong as to rule out values that might make sense. Point estimates are reported as medians with 95% quantile posterior intervals (QI). Descriptive model statistics are provided in S18 and S19 Tables in S2 File.

Finally, we report intervals around self-reported use, prevalence and perceived prevalence estimates (Figs 2–4) using Bayesian highest probability densities (hpd) computed with R binom package (v.1.1–1) using Jeffrey's prior, $Beta(\alpha, \beta = 0.5, 0.5)$. Jeffrey's prior is a non-informative prior that ensures the posterior distribution is invariant under reparameterisation. Additionally, it naturally accommodates the variability inherent in estimating proportions, leading to more robust and reliable posterior estimates, especially in cases with limited or sparse data. As we examine 25 different QRPs, we prefer this more objective approach, providing conservative estimates and ensuring our results are data-driven.

## Results

In the following, we present mean results for the 25 individual QRP statements (Figs 2–4) and results from models predicting self-reported engagements with QRPs (Figs 5–7).

In total, 30,618 QRP statements were presented in the Danish survey, 29,052 were completed, and 10,841 (37%) were reported as having been used. The corresponding numbers from the international survey are 11,772 QRP statements posed, 11,264 completed, and 4,534,

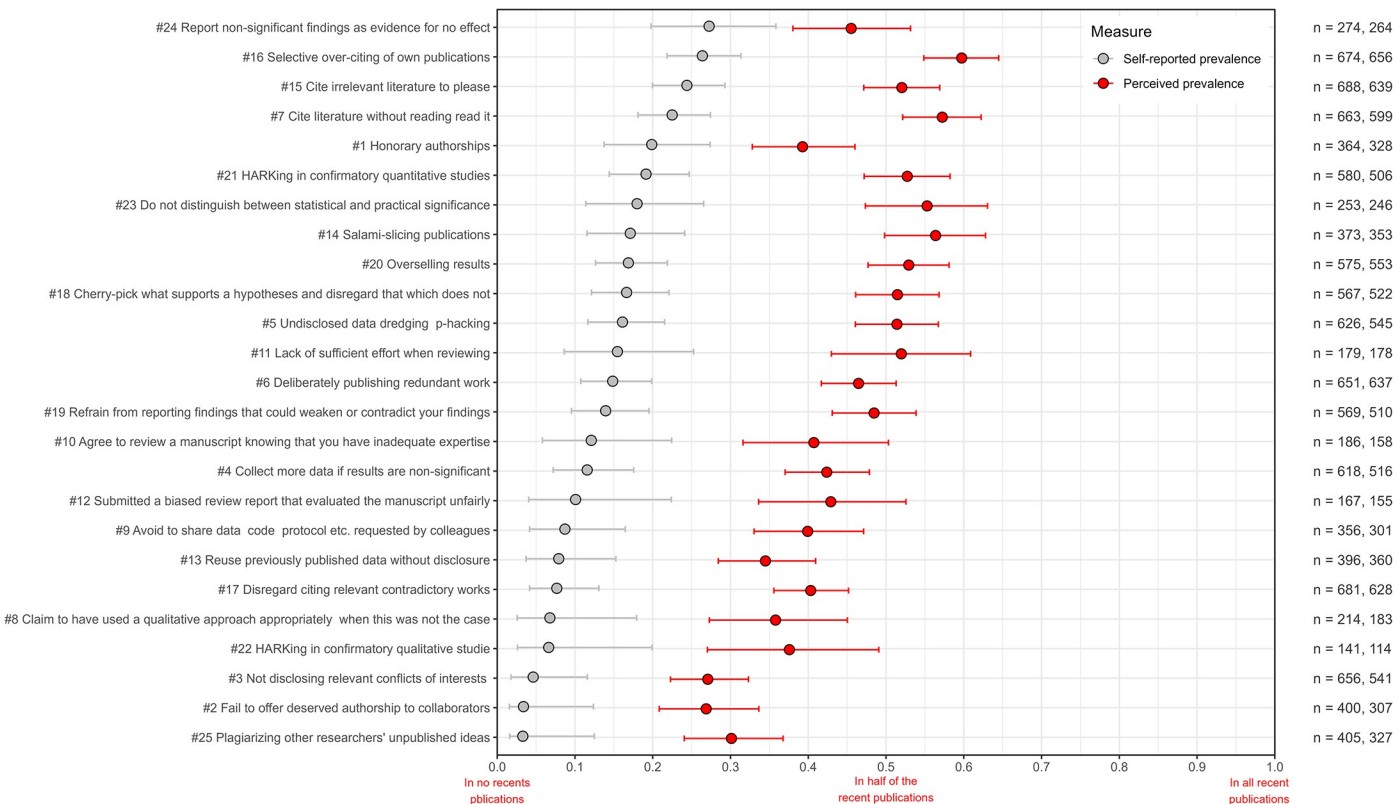

**Fig 4. Estimates of the self-reported (grey) and perceived (red) prevalence of QRPs in recent publications from the international survey.** QRPs are ranked according to self-reported prevalence. Proportions are reported with 95% Bayesian highest probability densities. The numbers of respondents (n) for each QRP are shown to the right, self-reported and perceived prevalence, respectively.

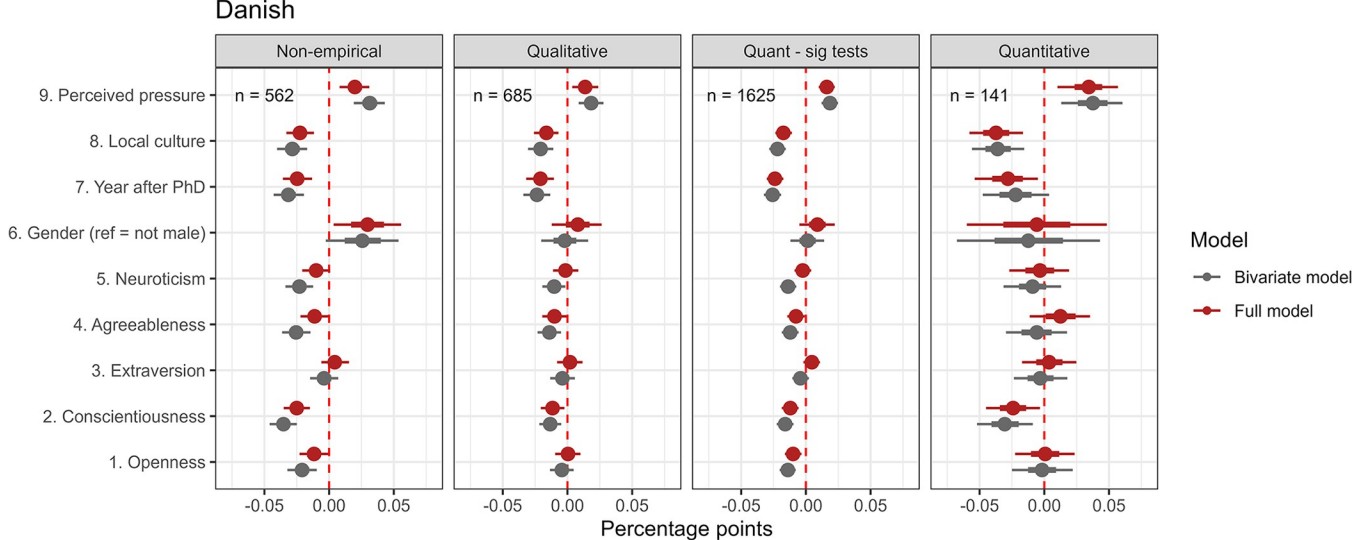

**Fig 5. Predictors of self-reported prevalence: Danish survey.** Response patterns across the four research approaches. Bayesian Linear Probability Model reported with medians and quantile intervals 66–95% Credible Intervals.

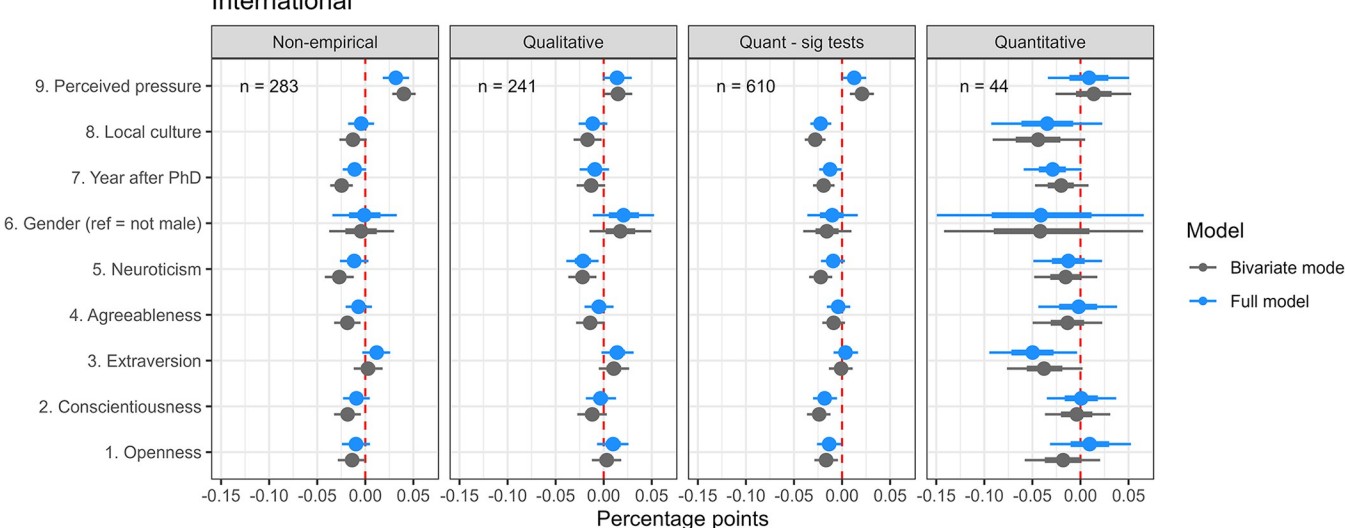

**Fig 6. Predictors of self-reported prevalence: International survey.** Response patterns across the four research approaches. Bayesian Linear Probability Model reported with medians and quantile intervals 66–95% Credible Intervals.

or 40%, were reported as having been used. Table 4 provides some basic response characteristics according to preferred research approaches. We see very similar response patterns across the two surveys. More than 9 out of 10 respondents answered seven or more statements for all approaches. We see a slightly higher median number of admitted statements for the approach 'quantitative in fields using significance test' among respondents in the international survey. Otherwise, the median for all other approaches is stable at three admitted statements.

## Self-reported use of 25 individual QRPs

Fig 2 presents the mean proportions of respondents from the Danish and international surveys who admitted use of a QRP at least once in recent publications (see S20-S23 Tables in S2 File for supplementary statistics and S3 File for a comprehensive summary of surveys we compare our findings to below). The potential QRP statements are ranked according to the frequency in the Danish set where 'selective over-citing of own publications' is the most frequently admitted, and 'plagiarising other researchers' unpublished ideas' is the least admitted practice.

**Table 4. Response characteristics according to preferred research approach.**

| Survey | Research approach | Statements answered | | | Statements admitted | |
|---|---|---|---|---|---|---|
| | | 9 | 7 or more | Mean | Median | Mean |
| Danish | | | | | | |
| n = 629 | Non-empirical | 82% | 98% | 8.7 | 3 | 3.1 |
| n = 797 | Qualitative | 48% | 95% | 8.3 | 3 | 2.9 |
| n = 160 | Quantitative | 77% | 96% | 8.6 | 3 | 3.3 |
| n = 1,816 | Quantitative in fields using significance tests | 79% | 96% | 8.6 | 3 | 3.3 |
| International | | | | | | |
| n = 325 | Non-empirical | 86% | 98% | 8.7 | 3 | 3.1 |
| n = 270 | Qualitative | 52% | 94% | 8.2 | 3 | 3.1 |
| n = 48 | Quantitative | 94% | 100% | 8.9 | 3 | 3.6 |
| n = 664 | Quantitative in fields using significance tests | 72% | 96% | 8.6 | 4 | 4.3 |

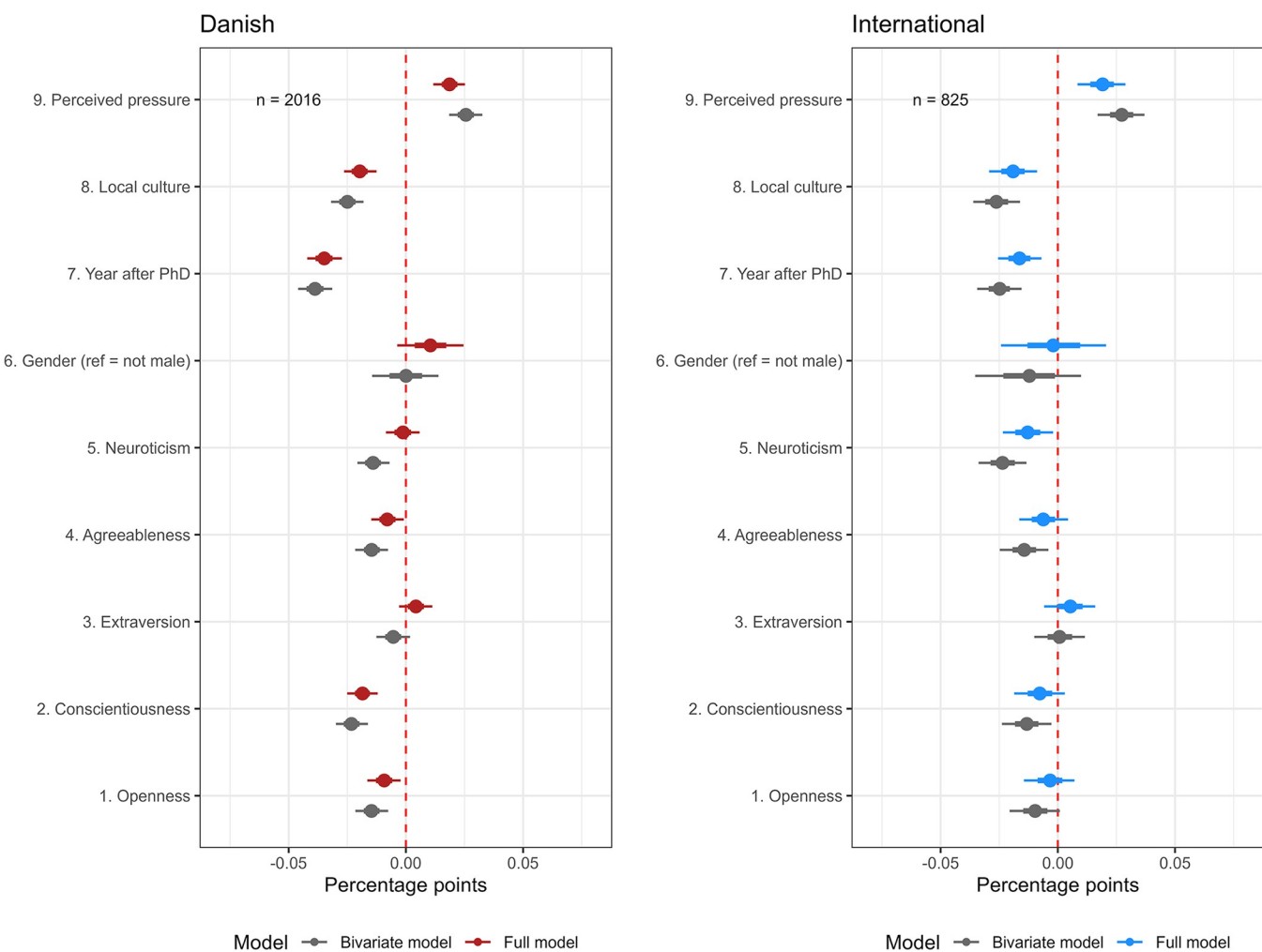

**Fig 7. Predictors of self-reported prevalence: Comparison of the Danish and international response patterns based on a subset of 11 QRP statements eligible to all respondents.** Bayesian Linear Probability Model reported with medians and quantile intervals 66–95% Credible Intervals.

We will focus on the Danish survey (DK) findings and compare response patterns to the international survey (INT). QRP numbers are indicated with a hashtag plus a number in the text and the figures.

Among the Danish respondents, self-reported use of specific QRPs ranged from 10 to 64%; in the international survey, the range was 11 to 65%. Other comparable surveys also reported ranges from 65–67% down to 5%, albeit usually for more restricted sets of QRPs.

The rank-order correlation of QRPs between the two surveys was very strong ($\tau = 0.85$). Roughly three strata emerged from the Danish rank-ordered set. The most frequently admitted practices used in recent publications comprised 14 QRPs, spanning from 64 down to 38% mean self-reported rates. A middle group of eight QRPs ranging from 30 down to 19%, and finally, a small group of three QRPs with the lowest self-reported use between 10 and 11%. Interestingly, the three most admitted uses of QRPs in both surveys are all potentially questionable citing practices: 'Selective over-citing' (#16) (DK 64%, INT 65%), 'cite without reading' (#7) (DK 59%, INT 60%), and 'irrelevant citing to please' (#15) (DK 59%, INT 62%). The only comparable surveys that also reported questionable citing practices found that among health professions education researchers (Artino et al. [12]), 50% admitted that they had cited

to please and cited without reading, and 30% admitted that they had selectively over-cited their works. Surveying European economists, Necker [50] also found questionable citing practices among the most readily admitted; 52% of the respondents cited without reading, and 59% cited strategically to please. The fourth statement on questionable citing practices, 'disregarding citing relevant contradictory works' (#17), is admitted considerably lower (DK 23%, INT 24%). Necker [50] also found considerably lower self-reported admittance for a similar item, 21% refrained from citing contradictory results or opinions.

The present survey is among the first to report on potential questionable reviewing practices where two of three statements included are among the most readily admitted: 'Not putting enough effort into reviewing' (#11) (DK 54%, INT 48%) and 'reviewing manuscripts despite having inadequate expertise to do so' (#10) (DK 38%, INT 37%). A third statement admitted that the use of 'unfair review reports' (#12) was reported somewhat lower (DK 26%, INT 27%). Also among the most admitted uses in recent publications were two statements linked to misuses or misunderstandings of significance tests, which are usually not included among the canon of QRPs targeting such procedures: 'Reporting non-significant findings as evidence for no effect' (#24) (DK 53%, INT 55%) and 'not distinguishing between statistical and practical significance' (#23) (DK 52%, INT 48%). Some would argue that these are not genuine QRPs. Nevertheless, both practices are part of the 'null hypothesis significance testing' (NHST) ritual [84] and have for decades been labelled as detrimental, the former even a logical fallacy [e.g., 85–87], yet seemingly to minimal effect. Failures to comprehend the flaws of NHST cause continuous misuse or misunderstanding of these questionable practices, which can be labelled as 'misleading interpretations'. Further, in this group of most frequently admitted QRPs are several well-known practices from comparable studies, albeit worded somewhat differently in this present questionnaire. In essence, these practices address issues of undisclosed selectivity. Selectivity in data analyses such as 'HARKing' (#21) (DK 47%, INT 49%) or 'p-hacking' in quantitative studies (#5) (DK 38%, INT 43%), or selectivity in reporting, such as 'cherry-picking what supports a hypothesis and disregard that which does not' (#18) (DK 41%, INT 48%). Previous comparable findings report HARKing between 27 and 54%, various kinds of p-hacking between 24 and 62% and selective reporting between 16 and 67% depending on specific practices surveyed (e.g. studies by Fraser et al. [13], Agnoli et al. [62], John et al. [65], Makel et al. [67], Bakker et al. [69] and Latan et al. [71]).

Another form of misleading is 'spin' or 'overselling' of results, a QRP not previously surveyed. Respectively, 45% (DK) and 56% (INT) admitted 'overselling' results (#20) at least once in recent publications. A couple of questionable publishing practices often categorised as 'recycling' and linked to research 'waste' were also frequently admitted, so-called 'salami-slicing' of publications (#14) (DK 44%, INT 46%) and 'deliberate publishing of redundant work' (#6) (DK 40%, INT 45%). In domain-specific surveys (Agnoli et al. [12], Necker [50]), 'salami-slicing' is reported at 22% and 20%, respectively. Finally, in this group of most frequently admitted practices, we find that 'honorary authorships' (#1) were self-reported at 49% (DK) and 45% (INT). Artino and colleagues [12] reported 'honorary authorships' at 61%. Older surveys in medicine reported 'honorary authorships' around 19% (e.g., Flanagin et al. [88]), in line with the meta-analysis by Xie and colleagues [60]. In contrast, later surveys reported considerably higher estimates, 55% (Rajasekaran et al. [89]), which is in line with the documented inflationary developments in co-authorships in recent decades (Persson et al. [90], Fanelli & Larivière [91]).

In the middle group, besides the already mentioned statements above about citing and reviewing practices (#17, #12), we found further instances of undisclosed selectivity in reporting, such as 'refrain from reporting findings that could weaken or contradict your findings' (#19) (DK 30%, INT 41%), and concerning analyses, 'collect more data if results are non-

significant' (i.e. also p-hacking) (#4) (DK 23%, INT 29%), or the novel statement 'HARKing in confirmatory qualitative studies' (#22) (DK 26%, INT 23%) which was reported at a considerable lower level than the quantitative twin. The former two QRPs, albeit somewhat differently formulated, were among the canonical items stemming from (John et al. [65]) and repeated in several subsequent surveys (e.g., Fraser et al. [13], Agnoli et al. [62], Fiedler & Schwarz [63], John et al. [65], Rabelo et al. [68], Chin et al. [70]). Estimates for both QRPs varied considerably over these surveys, from around 15 to 57%. Noticeably, higher estimates for 'collect more data' are typically linked to experimental fields. Related practices of secrecy or lack of transparency such as 'avoid to share data, code, protocols etc.' (#9) (DK 22%, INT 27%); 'claiming to have used a qualitative approach appropriately, when this was not the case' (#8) (DK 20%, INT 21%); and 'reuse previously published data without disclosure' (#13) which also taps into recycling, was self-reported at 19% in the Danish survey and 25% in the international survey.

Finally, the small group of three statements with the lowest self-reported use in the Danish survey include another authorship issue: 'Failing to offer deserved authorship to collaborators' (#2) (DK 11%, INT 12%). Compared to other questionable authorship practices, the use of this one was markedly lower. This is also the case in (Necker [50]) where a similar item was reported at 1.5%, or in (Artino et al. [12]) at 6%. The two final statements address a transparency issue of 'not disclosing relevant conflicts of interests' (#3) (DK 11%, INT 12%). Finally, 'plagiarising other researchers' unpublished ideas' (#25) would, in many settings, be perceived as misconduct rather than a questionable practice. Indeed, the severity of this action may be indicated by the respondents in both surveys, who self-reported use at 10% (DK) and 11% (INT), respectively, which is the lowest in both data sets.

Consequently, patterns of self-reported use of the 25 QRPs were also remarkably similar in our two surveys. Where comparable, our findings are generally also in line with those from other surveys (e.g., Artino et al. [12], Fraser et al. [13], Necker [50], Agnoli et al. [62], John et al. [65], Makel et al. [67], Bakker et al. [69], Chin et al. [70], Latan et al. [71]). It seems evident that neither country nor field of research particularly influences respondents' motivation to self-report the use of QRPs, but there are indications that willingness to admit is linked to the perceived social acceptability of a practice The rankings in Fig 3 cluster practices which are presumably seen by many as trivial breaches of integrity, if breaches at all, such as questionable citing practice, on top and plagiarism at the bottom. We explore this presumed clustering of social acceptability below.

## Self-reported and perceived prevalence of 25 individual QRPs

Next, we examine prevalence according to Fiedler and Schwarz [63]. We present self-reported and perceived prevalence together, albeit separately, for the two surveys, Fig 3 (DK) and Fig 4 (INT). Consider 'selective over-citing of own publications'. This is the QRP statement that most respondents admitted to using at least once in recent publications, 64% (DK) and 65% (INT), respectively. Self-reported prevalence estimates, however, were 26% in both surveys, ranking the statement second highest when it comes to prevalence while first when it comes to admitted use. The most prevalent QRP in both surveys was 'report non-significant findings as evidence for no effect'. While 53% (DK) and 55% (INT) admit use in recent publications, the resulting prevalence is 26% (DK) and 27% (INT), or roughly one in four recent publications. So, half of the respondents receiving this QRP statement admitted use in recent publications. Yet, they did not necessarily indicate frequent, systematic use according to the self-reported scale. According to Fiedler and Schwarz [63], the latter is a genuine estimate of prevalence that multiplies the average admitted use by the average frequency with which use is reported on some scale. Consequently, while two-thirds of the respondents admitted recent use, the

intensity by which they used it suggests that one in four recent publications from our respondents have issues with 'selective over-citing'.

We should expect a strong rank correlation between self-reported use and prevalence, which indeed was the case ($\tau$ = 0.85 DK; 0.88 INT). The most prevalent QRP in both surveys was 'report non-significant findings as evidence for no effect' (#24) (DK: 26%, INT: 27%), which also had the highest frequency rates. What caused the changes in the prevalence rankings compared to self-reported use was the varying frequency rates with which QRPs were reported. Among our respondents, roughly one in four recent publications using significance tests 'reported non-significant findings as evidence for no effect'. On average, respondents admit more intensive use of this QRP compared to all other QRP statements, which indicates that this practice is seen by many as normative or perhaps less problematic. Either respondents resonate that this is what we and others do, not realising the intricacies of interpreting 'significance tests', and that they, in this case, make faulty claims, or they do not acknowledge that such a practice is questionable. Indeed, some would argue that it was even detrimental [92], albeit still making a faulty claim.

We will not go through the 25 QRP statements but instead emphasise the conceptual differences between the two quantities of self-reported use and prevalence. Fiedler and Schwarz [63] report that across all their items, self-reported use rates were about five times higher than their prevalence estimates. Our set of QRP statements is much more diverse, and we find smaller discrepancies. Self-reported use was, on average, 'only' three times higher than prevalence in the Danish survey and 2.9 times higher in the international survey. So for clarity, one out of two respondents in the Danish survey admitted recent use of 'honorary authorships', but the prevalence estimate suggests that the half who admitted having engaged with 'honorary authorships' recently, on average, did it in one out of five of their recent publications, or more generally interpreted, 20% of recent publications from the respondents include 'honorary authorships'.

To what extent did respondents believe that QRPs are common in the speciality literature they are involved in? Perceived prevalence estimates are less interesting, but when compared with self-reported patterns, they are often assumed to provide insights into social acceptance of research practices. Consequently, perceptions of widespread use and the gap between perceived and self-reported prevalence may indicate to what extent respondents implicitly see practices as socially acceptable or perhaps more simply as a way of vindicating their use.

Suppose response patterns should indicate the social acceptability of QRPs. In that case, we should expect strong rank correlations between self-reported and perceived prevalence estimates, which indeed were the cases ($\tau$ = 0.77 (DK), 0.68 (INT)). Consequently, when self-reporting becomes more frequent, so does the perception of prevalence in recent literature, albeit with some notable deviations. On average, for all QRPs, the perceived prevalence was 3.9 times higher than the self-reported prevalence in both surveys. IQRs were 2.8–4.7 (DK) and 3.1–4.6 (INT), and medians were 3.5 (DK) and 3.4 (INT). The general pattern in both surveys was that respondents perceived the six most prevalent QRPs as 2–3 times more prevalent in the literature than their self-reporting rates. The ranks from 7 to around 16, 3–4 times more prevalent; ranks from 17 to 19, 4–5 times more prevalent; ranks from 20 to 24, 5–6 times more prevalent, although (#2) in the international survey was an outlier, eight times more prevalent; and finally, (#25) at the bottom rank, perceived to be 9–10 times more prevalent compared to the self-reported prevalence estimates. As self-reported prevalence estimates increase, the ratios with perceived prevalence eventually decrease somewhat and vice versa. Nevertheless, the overall response patterns open up for interpretations of implicit hierarchies of social acceptability. There are some notable examples.

'Plagiarising other researchers' unpublished ideas' (#25) can be seen as the most severe among the 25 as it addresses a form of misconduct. This QRP has the lowest self-reported prevalence estimates in both surveys, which was expected due to its presumed severity. It is unclear, however, why it is perceived to be more prevalent than QRP2 and QRP3. Nevertheless, the low self-reported prevalence estimates and the distinct gap to the perceived estimates 9–10 times higher suggest a low social acceptance of plagiarising. Conversely, when we look at the top-ranked QRP statements, such as 'report non-significant findings as evidence for no effect' (#24) or 'cite irrelevant literature to please' (#15) or 'honorary authorships' (#1), we find the lowest gaps between self-reported and perceived prevalence—a clear indication of more perceived social acceptability.

As with other general findings, we also see similar response patterns regarding perceived prevalence in the two surveys, $\tau = 0.85$, and mean perceived prevalence for all QRPs are 40% (DK) and 45% (INT).

## Predictors of prevalence

Finally, we examined potential predictors of self-reported prevalence, factors that may cultivate or limit engagement with QRPs. We focused on a general model that incorporated nine predictors. The outcome variable was respondents' self-reported prevalence, which expresses the frequency with which they admitted engagements with QRPs in recent publications. We modelled each of the four preferred research approaches separately and for each survey. Results are presented in Fig 5 (DK) and Fig 6 (INT) (S24 Fig in S2 File presents results from a model where responses from the two surveys are combined). Additionally, for each survey, we combined all respondents in a model using the subset of 11 QRPs applicable to everyone (Fig 7). The figures present partial coefficient estimates and their bivariate variants for each predictor. Note that the results from the Danish survey are more robust due to its sample size being roughly three times larger than that of the international survey. Therefore, we primarily refer to the Danish survey when comparing the two. We are mainly interested in the extent to which patterns are similar between the surveys, acknowledging that the uncertainty in the international data set is greater.

First, we will look at the results of the Danish survey (Fig 5). The median posterior estimates of QRP prevalence for respondents eligible for the four preferred research approaches were: non-empirical: 11% QI [8.6, 13.5], $n = 562$; qualitative: 11.5% QI [10.3, 12.8], $n = 685$; quantitative (sig test): 12.9% QI [11.8, 13.9], $n = 1,625$; and quantitative: 14.2% QI [9.4, 18.9], $n = 141$ (i.e. intercept models). Corresponding results from the international survey were non-empirical: 13.2% QI [10.4, 15.9], $n = 283$; qualitative: 12.2% QI [10, 14.4], $n = 241$; quantitative (sig test): 17.3% QI [15.5, 19.2], $n = 610$; and quantitative: 16.8% QI [7.2, 26.5], $n = 44$ (i.e. intercept models). Medians were slightly lower in the Danish survey, albeit the patterns between groups seemed similar for the two surveys.

Generally, the sizes of the estimated partial coefficients were small. Not surprisingly, the most robust positive predictor of prevalence in both surveys was 'perceived pressure'. In the Danish survey, one z-score increase, increased prevalence respectively by 2, QI [0.8, 3.1] percentage points for non-empirical, 1.4, QI [0.4, 2.4] for qualitative, 1.6, QI [1, 2.2] for quantitative (sig test), and 3.4, QI [1, 5.7] for quantitative. In Bayesian terms for the non-empirical approach, the quantile interval suggests that, based on our model and prior information, there is a 95% probability that the parameter lies between 0.8 and 3.1 percentage points, with the highest probability density around the median 2 points.

Gender seems to be a positive predictor across at least three of the four research approaches in the Danish survey (non-empirical: 3, QI [0.4, 5.6]; qualitative: 0.8, QI [-1.2, 2.6]; and

quantitative (sig test): 0.9, QI [-0.5, 2.2]); i.e. the masses of the posterior probability distributions are generally positive) male respondents report slightly higher QRP prevalence rates than non-male respondents. However, this is not the case in the international survey, where results are more mixed across the research approaches.

Contrary to several other surveys (Fraser et al. [13], Chin et al. [70]), we found that academic age (year after PhD) was a robust negative predictor for engagements with QRPs in both surveys (DK: non-empirical: -2.5, QI [-3.6, -1.3]; qualitative': -2.1, QI [-3.2, -1.1]; quantitative (sig test): -2.4, QI [-3.0, -1.7]; and quantitative: -2.8, QI [-5.4, -0.5]). The average ages across the research approaches were 11 to 13 years after PhD in the Danish survey and 15 to 17 years in the international survey, with standard deviations of 9 and 12 years. The negative association suggests that respondents with longer careers behind them self-report less engagement with QRPs.

Besides academic age, 'local research culture' (DK: non-empirical: -2.2, QI [-3.3, -1.2]; qualitative: -1.6, QI [-2.6, -0.7]; quantitative (sig test): -1.7, QI [-2.4, -1.1]; 'quantitative': -3.7, QI [-5.8, -1.6]) and conscientiousness (DK: non-empirical: -2.5, QI [-3.5, -1.5]; qualitative: -1.2, QI [-2.1, -0.2]; quantitative (sig test): -1.2, QI [-1.9, -0.6]; quantitative: -2.4, QI [-4.5, -0.3]) were the strongest negative predictors of QRP prevalence across the research approaches in the Danish survey. 'Local research culture' and, to a lesser extent, conscientiousness are also negative predictors in the international survey. These patterns were also applied to a basic model where the five personality traits were omitted. When included, conscientiousness, as expected, showed the strongest negative relations with self-reported prevalence among the five individual factors. To a lesser degree and with some variation across the research approaches, openness, agreeableness, and neuroticism were also slightly negatively associated with self-reported prevalence, whereas extraversion showed minuscule associations in both surveys.

Finally, Fig 7 shows full and bivariate models for each survey using the subset of 11 QRPs available to all respondents) (S25 Fig in S2 File presents results from a model where responses from the two surveys are combined). We use this model to examine the robustness of the patterns discerned from the previous models. Considering that most respondents were presented with fewer than nine of these QRPs, only respondents presented with three or more of these QRPs were included to ensure more robust estimates. We consider the restricted samples well-balanced, but there are some changes in the relative composition of respondents compared to the overall composition of the total samples across the preferred research approaches (see Table 2). The restricted samples are respectively $n = 2016$ (DK) and $n = 825$ (INT), and the new relative compositions of respondents are: non-empirical (DK: 28%; INT: 35%); qualitative (DK: 29%; INT: 25%); quantitative (sig test) (DK: 36%; INT: 35%); quantitative (DK: 5%; INT: 4%). The most marked change is that respondents from the quantitative (sig test) groups, while still the largest, now only constitute some 36% (DK) and 35% (INT) of respondents, while in the total samples, they constituted 53% and 51%, respectively.

The primary response patterns were reproduced in these two models. 'Perceived pressure' was again the strongest positive predictor of engagement with QRPs in both models, 1.9, QI [1.2, 2.5] (DK) and 1.9, QI [0.8, 2.9] (INT). Likewise, 'local culture', -2.0, QI, [-2.6, -1.3] (DK) and -1.9, QI [-2.9, -0.9] (INT), and 'academic age' -3.5, QI, [-4.2, -2.7] (DK) and -1.6, QI [-2.6, -0.7] (INT), were the strongest negative predictors. Conscientiousness (-1.8, QI, [-2.5, -1.2]) remains a relatively strong negative predictor of engagement for the Danish set. Likewise, openness (-0.9, QI, [-1.6, -0.2]) and agreeableness (-0.8, QI, [-1.5, -0.1]) are also still negatively associated with self-reported prevalence. Gender also seems positively associated with engagements (1.0, QI [-0.4, 2.5]) as 66% of the posterior distribution is positive, and the median is 1. These patterns are not so markedly discernible from the international set. Conscientiousness is most likely also a negative predictor but to a lesser degree (-0.8, QI [-1.9, 0.3]). Openness (-0.3,

QI [-1.4, 0.7]) and agreeableness (-0.6, QI [-1.6, 0.4]) are more unclear, which is also the case for Gender (-0.2, QI [-2.4, 2.1]). However, the results from the international set deviate from one predictor, neuroticism (-1.3, QI [-2.3, -0.2]), which turns out to be negatively associated with engagement. The other predictors showed some variations between fields but characteristically with broader posterior probability intervals.

## Discussion

The present survey results suggest that perception, use and prevalence of QRPs among participating researchers in Denmark do not differ from the response patterns of researchers working in the UK, USA, Austria and Croatia, so figuratively speaking, something does not seem to be 'rotten in the state of Denmark'. However, overall results from the two surveys suggest common, widespread involvement with QRPs in recent publications, seemingly supporting the narrative that 'science is facing systemic problems'. However, findings also suggest that not everyone is equally likely to succumb to problematic system-wide incentives.

The overall response patterns are in line with many previous comparable surveys claiming widespread use of QRPs (e.g., Artino et al. [12], Necker [50], Agnoli et al. [62], Fiedler & Schwarz [63], John et al. [65], Bakker et al. [69]; Chin et al. [70]). They are also in line with the pooled estimate obtained by Fanelli [61] but much higher than the recent estimate from Xie et al. [60]. Self-reported use of the 25 specific QRPs ranged from 10–64% (DK) and 11–65% (INT), where the rank order of QRPs between the two surveys was very strong ($\tau = 0.85$). While our set of QRPs is much more varied, other surveys also reported ranges from 67% down to 5%, indicating stable response patterns across surveys irrespective of QRPs.

On average, we found self-reported prevalence rates roughly three times lower than self-reported use rates. Prevalence estimates ranged from 3% to 26–27% in the two surveys. An intriguing question is what would be perceived as prevalent in the sense that its rate of use is worrying. Fiedler and Schwarz [63] themselves argued that "[t]he most unpleasant result of our attempt to disambiguate survey data on scientific norm violations is that their prevalence rate is not zero!" From this yardstick, one in four publications may seem prevalent and worrying.

Fiedler and Schwarz [63] criticised John and colleagues [65] for using the concept of prevalence when, in fact, they reported use and that this generally has led to inflated prevalence estimates propagating a descriptive norm that QRPs are highly prevalent. We acknowledge the conceptual differences between use and prevalence and admit the challenges in their interpretations, but based on the present findings, and in line with previous ones, it seems appropriate to conclude that most researchers will admit use of at least one QRP in a set of statements given to them and on average they will admit around three when given nine such statements. This supports a narrative of widespread involvement in QRPs' of all sorts among researchers. However, general prevalence estimates also suggest the low frequency of use among individual researchers, which provides more limited evidence for claims of systematic use of particular QRPs. Some recent surveys have also tried to address the frequency of use but in different ways (Fraser et al. [13], Gopalakrishna et al. [58], Latan et al. [71]). While these operationalisations of prevalence are different, they all suggest one important lesson. When respondents grade their use on a scale, most will admit to using one or several QRPs, but they will most likely indicate low frequency or non-systematic use, resulting in distinct right-skewed distributions (see S23 Table in S2 File). To what extent does this reflect valid estimates, or is it more a consequence of social desirability? It is challenging to ascertain.

Like others, we also find merit in comparing self-reported and perceived prevalence patterns, assuming that they can provide crude insights into social acceptance of research

practices, perhaps even normative use, or at least suggest vindications for own use. Overall, we did find strong rank correlations between self-reported and perceived prevalence patterns in both surveys. The varying gaps between the rank-ordered pairs provide intuitive albeit crude and only suggestive indications of different degrees of social acceptability and normative perceptions.

Chin and colleagues [70] have recently argued that mean-perceived prevalence estimates can be misleading if the underlying distribution of participants' responses resembles a uniform distribution, creating mean estimates of around 50%. According to Chin et al. [70], such a response pattern suggests that most participants know very little about the prevalence of these practices in their field and that descriptive norms about research behaviour may be weak and only weakly tied to reality. We do not find strong indications of uniform distributions in the present surveys. However, we acknowledge that guestimates of perceived prevalence are ambiguous and influenced by the scale and its anchors, but most importantly, presumably also the perceived social acceptability of a QRP. The latter is what makes such analyses informative in our view. Response patterns of perceived prevalence in comparison with self-reported prevalence inform us of perceptions of social acceptability, normative use, and perhaps vindication strategies, and in that sense, these patterns resemble what Anderson and colleagues [27] in their study of norms of behaviour in scientific research call the 'candour/conceit gap', i.e., researchers admit their behaviour falls short of ideals, but believe that it comes closer than their colleagues' behaviour. It seems that self-reported surveys on integrity issues in research in general document such response behaviours.

Finally, our modelling of potential predictors of self-reported prevalence suggests, in line with previous studies, that the systemic factor 'perceived pressure' is the strongest predictor for self-reported prevalence (e.g., Gopalakrishna et al. [58]). This could indicate that the prevalence of QRP engagement, to some extent, is associated with researchers' perceptions of expected performance in the 'hyper-competition' they are engaged in, and thus, to some extent, supports the narrative about 'corrupting' systemic structures in science (e.g., Edwards & Roy [38], Boyle & Götz [93], Giner-Sorolla [94]). A more subtle conjecture could be that these correlations, to some extent, also represent vindications, blaming the system's perverse incentives for one's questionable practices—"it's the incentives" (Yarkoni [33])! Conversely, the institutional factor 'local culture' has a negative association with self-reported prevalence, which suggests that stronger local focus on leadership and research cultures that promote quality, rigorous research, and reward integrity, to some extent, seem to counter systematic use of QRPs despite presumed systemic challenges. This finding aligns with Gopalakrishna et al. [58], which suggests strengthening emphasis on scientific norm subdescription. Contrary to other surveys, we do find that respondents with longer careers behind them self-report less engagement with QRPs. It is, however, uncertain to what extent this is an actual manifestation of behaviour or rather an expression of perceptions that one does not violate norms. It is important to note that these systemic relationships may be inflated to some degree from reverse causality. The perception of being pressured by external forces, for example, may not only serve as a potential cause of questionable behaviour but also as an excuse, as suggested by studies on cognitive dissonance and motivated reasoning (Kunda [95]).

Based on relatively weak associations, Tijdink and colleagues [37] claim that Machiavellianism may be a risk factor for misbehaviour and suggest that personality impacts research behaviour and should be taken into account in fostering responsible conduct of research. This aligns with the 'bad apple' narrative of misbehaviour. Somewhat different, our overall findings align with Giluk and Postlethwaite [42]. Their meta-analysis of personality traits and academic dishonesty claims that conscientiousness and agreeableness are the strongest predictors negatively related to academic dishonesty. Conscientiousness is the tendency to be organised, goal-

directed, to delay gratification, and to follow norms and rules (Roberts et al. [96]). Agreeableness is concerned with how individuals approach interpersonal relationships. Agreeable individuals are likeable, helpful, trusting, and concerned with the welfare of others. Indeed, these traits align with notions of responsible conduct research, such as honesty and respect (e.g. the European Code of Conduct for Research Integrity [97]). Low conscientiousness may, therefore, be the most relevant trait in relation to 'academic dishonesty' because it has the closest conceptual connection to cheating (Williams et al. [98]). Our findings, especially from the Danish survey, suggest that respondents scoring higher on conscientiousness and agreeableness will tend to report less intensive engagement with QRPs, suggesting they are less willing to 'cut corners' intentionally.

Consequently, the two surveys have similar main patterns with few exceptions. The fact that 'local research culture', 'academic age' and conscientiousness are the strongest negative predictors in our two surveys suggests that explanations of self-reported perception and prevalence of QRPs are a complicated mixture of experience, systemic features, and individual traits, but perhaps also motivated reasoning.

## Limitations

Surveys trying to quantify the occurrence of QRPs using self-reported data are not ideal, and neither is the present study. Response rates of 22% (DK) and 4% (INT) are not impressive. While the former is among the highest for comparable studies, it seems futile to defend it as representative of the population of researchers in Denmark when it is almost certain that self-selection is an issue. We did implement a list experiment in the questionnaire, but unfortunately, the implementation of the instrument did not work out as planned (see also Jerke et al. [99]), so we do not have a measure of possible social desirability in the self-reported response patterns. Nevertheless, we have two surveys that differ considerably in response rates, are biased by self-selection and social desirability to some unknown extent and show remarkably similar response patterns.

An important challenge turned out to be the measurement of prevalence itself. In retrospect, asking respondents to grade their intensity of use on a 7-point scale framed in relation to recent publications may have been a too complex cognitive task. Respondents may grasp whether they have used a particular QRP recently. Still, we are inclined to believe that judging their intensity of use is challenged by memory and social desirability issues that lead to response patterns where most of the scoring will be in the lowest non-zero categories of the scale. When we examine distributions of responses on ordinal scales in the present and other comparable surveys (Fraser et al. [13], Gopalakrishna et al. [58], Fiedler & Schwarz [63]), we identify such highly right-skewed distributions. Such response patterns largely dictate prevalence measures used here and elsewhere [e.g., 58]. However, while response patterns seem very stable across studies and may suggest robust use and prevalence estimates, we suspect that they, to some unknown degrees, are inflated due to generic response behaviours. However, it is not certain that a simpler instrument would have altered the highly skewed response patterns. It may well be that when respondents admit QRP use on a scale, they overwhelmingly respond in the lowest categories irrespective of the length of the scale.

Another challenge is related to interpretations of prevalence in recent publications. We asked about practices in recent publications, and respondents presumably self-reported their use, most likely not what they knew or presumed about the practices of their potential co-authors. It is, therefore, not obvious that estimates derived from self-reporting can indicate prevalence in the literature beyond single-authored works. As a unit of analysis, it therefore seems that researchers and their practices may be more straightforward to interpret.

## Implications

Despite these limitations, our results contribute to the broader understanding of QRPs in several important ways.

First, using a different framing and a more extensive and much more diverse set of QRPs, we find remarkably similar response patterns in the Danish and international surveys, and most importantly, despite our different approaches, response patterns align with many other survey studies. The range and means of self-reported use of individual QRPs also align with boundaries found in previous studies. We, therefore, argue that these response patterns are very robust and seemingly reproducible despite different settings. They are, thus, probably 'as good as it gets' from such surveys of QRPs. Note that we do not claim these are 'unbiased' estimates; they are just reliable response patterns given the instrument.

Second, we have provided some much-needed conceptual clarifications on use and prevalence and documented their different quantities and interpretations. We find that self-reported use is probably the least problematic quantity to measure. Despite much confusion, use should be interpreted as the willingness to admit involvement with a practice. Findings across studies suggest that most respondents will admit engagement with some QRPs, and many admit using more than one, suggesting 'widespread use' among respondents.

While easy to quantify, self-reported prevalence is more elusive to measure due to possible biases in response patterns when judging one's intensity of use. It is clear, however, that prevalence will most likely produce quantities several times lower than self-reported use. Consequently, taken at face value, while prevalence rates vary among QRPs, most likely due to their degree of perceived 'acceptance', their considerably lower quantities do not suggest systematic use in recent publications for most QRPs. Many admit involvement but not systematic use. It is, therefore, important to stress that use and prevalence are different but complementary measures with different interpretations.

Third, our comprehensive study of a diverse set of potential QRPs tries to readjust the popular discourse on QRPs, which, despite definitions going back to at least the early 1990s [9], have been dominated by more recent narrow conceptions in experimental domains in the behavioural sciences framed almost entirely in relation to misuses of significance tests and resulting challenges of false-positive claims and reproducibility issues. While such practices are problematic and produce epistemic biases and probably unreliable literature where prevalent, they are not the only QRPs that may bias and, if prevalent, may corrupt the epistemic and social systems in science. Unfortunately, this constricted focus implies that the debate about potential causes and negative consequences of QRPs is restricted mainly to 'suboptimal use of the scientific method' (Hardwick et al. [100]), where the whole scholarly enterprise should be targeted. For example, in the context of QRPs, little attention has been drawn to the extent and consequences of questionable citing or reviewing practices. Among our respondents, such practices are among the most prevalently admitted and most likely also perceived as normative and not particularly severe. Ironically, 'normative' citing practice is usually ascribed to a view that sees citing as the reaffirmation of scientific behaviour's underlying virtue, hence 'giving credit where credit is due' (Kaplan [101]). In the context of QRPs, normative use is the opposite: perceptions of questionable citing practices in the literature and among peers. Extensive *p*-hacking and selective reporting have raised concerns about the credibility of knowledge claims. Similarly, extensive questionable citing and reviewing practices, as documented here, should cause more concern about vital elements of the scholarly process, to what extent they are biased and the potential consequences of such bias. Obviously, for decades, others have pointed to problems with peer review (e.g., Heesen & Bright [102]) and even suggested that the quality control system in science was probably broken due to the growth of the system itself (Ravetz [103]). Likewise, for decades, many studies have also argued that citing

practices and their derivative citations are flawed (see, e.g. Horbach et al. [23]). Still, this literature has mostly not been linked to integrity issues in general or QRP studies in particular. Our findings suggest prevalence and seemingly normative use in a manner that should be of concern to a system that relies on proper quality control systems, reliable dissemination of knowledge claims, prides itself on providing 'credit where credit is due' so that meritocracy is sustained, and insistently reward and promote its agents based on derived indicators of these practices.

Fourth, we examine and document that both systemic and individual factors correlate, albeit not very strongly, with the willingness to self-report QRP prevalence. As documented here and elsewhere, it seems beyond doubt that positive perceptions of 'being under pressure' to publish and get funding predicts more prevalent use of QRPs. Another issue is to what extent such 'pressure' is real, imagined, or even an 'excuse'. However, we also document that more active engagement with and fostering of 'research cultures' locally predicts a lower prevalence of QRPs. These important social factors can be addressed when trying to foster more responsible conduct of research. The latter is much more challenging when it comes to personality traits. What our surveys document is the association of traits that have previously been linked to 'cheating' or 'dishonesty' in academic settings (Giluk & Postlethwaite [42], Mazar, & Ariely [35]) are also predictors of self-reported QRP prevalence. Our findings suggest that respondents scoring higher on conscientiousness (opposite to unreliability) and agreeableness (opposite to hostility) will tend to report fewer intensive engagements with QRPs. Our findings suggest that the 'bad apple' narrative of misbehaviour does not suffice to partially explain widespread engagement in various QRPs. Personality traits indicate dispositions as they lower behaviour thresholds and make behaviours consistent with our most likely traits. Consequently, to varying degrees, systemic challenges are undoubtedly real and likely to influence motivated reasoning among scientists (May [34]). Still, our findings suggest that dispositions to negative impacts on behaviours from such challenges are not uniformly distributed across individuals. Consequently, individual-level engagements with QRPs seem to be somewhat associated with personality traits (Feist & Gorman [104]). This suggests that initiatives fostering responsible conduct of research should not overlook individual factors. Indeed, some research suggests that a widespread narrative that solely blames corrupt external factors may promote questionable behaviours because it provides an easy justification (Stone et al. [105]).

## Conclusions

Mazar and colleagues [35, 106] provocatively claim that dishonesty is not about 'bad apples' but is part of the 'human condition', suggesting that all researchers are dishonest, but only to a limited extent. Dishonest actions are presumably prevalent because researchers constantly encounter situations where they can gain advantages by cutting corners [35]. In other words, dishonesty pays off, and researchers behave dishonestly enough to profit from it, but only to the extent that they can delude themselves into maintaining their integrity [35]. We disagree with this strong claim. In our view, engaging in potential QRPs does not necessarily equate to consciously t cheating to gain advantages or being dishonest. A multitude of individual practices characterises research processes, each comprising numerous more or less considered choices and heuristics, many of which are associated with accepted norms, no matter how problematic they may seem to some. Thus, research output results from a long path of more or less conscious or well-considered choices and heuristics made alone or collectively among collaborators. We argue that many of these practices, although perhaps perceived as questionable or directly detrimental by some, are not unequivocally so and are not necessarily always carried out with the intention to deceive to gain advantages. Advantages can be gained from unintended or sloppy practices, which is one of the reasons why many QRPs create biases to some

extent, depending on their severity and prevalence combined. However, we argue that it is not necessarily about all researchers 'being dishonest to a limited extent'. It is more about all researchers, more or less consciously, moving within an 'ethical grey zone' of research behaviours during the research process. A zone where the lower boundary between good and questionable practices is unclear and where darker shades of grey indicate more severe practices that are less engaged in, presumably because one is aware of their severity and norm violation. Such practices are closer to the clearly defined upper boundary of misconduct, the black zone, which includes deliberate acts of fabrication, falsification, and plagiarism. We argue that researchers' behaviours do not reside at a fixed point on this continuum from good practices (white) through questionable practices (grey) to misconduct (black) but are in flux throughout the research process with its many choices, heuristics, and norms. Whether a specific practice in a particular context is questionable is indeed questionable—'it depends.' This is likely one of the reasons why we see claims of widespread use or involvement in QRPs. It also raises important questions about the nature of QRPs and to what extent it is meaningful to categorise individual practices as being inside or outside a discrete category. Many such practices cannot be unequivocally classified as questionable in the problematic sense. They are context-dependent. The view of them can change over time, and their degree of questionability and severity are subjects of perpetual discussions within and between research communities.

A publication is a stylised condensation of a complex research process, which includes many more or less deliberate choices, heuristics, more or less arbitrary defaults, and rules of thumb. QRPs appear to be an inevitable part of this process. QRPs are typically examined individually. We ask whether a researcher has used a specific practice. However, a publication contains a sum of such practices from all involved contributors—QRPs often occur in bundles (Gerrits et al. [107]). Most publications will, therefore, be based, to a greater or lesser extent, on more or less questionable practices done inadvertently and intentionally. Given their indeterminate nature, countering all of them seems to be a Herculean task, likely even an impossible one. However, this does not mean that research cultures, norms, and reward structures cannot or should not be improved. They should always be, although this is not an easy task, not least because of the internal resistance within the system itself from gatekeepers and powerful actors who benefit from the status quo (Schneider et al. [108]).

## Supporting information

**S1 Table. QRR statements, category, and wording in the questionnaire.**
(PDF)

**S1 File. Assignments of QRPs to preferred research approaches.**
(PDF)

**S2 File. Supplementary information for empirical analyses (S4, S10-S23 Tables and S5-S9, S24 and S25 Figs).**
(PDF)

**S3 File. Comprehensive summary of comparable surveys (S25-S30 Tables).**
(PDF)

## Acknowledgments

We thank Kaare Aagaard, Asger Dalsgaard Pedersen, Pernille Bak Pedersen (survey development), Allan Rye Lyngs (web scraping), Ana Marusic, Nicole Foeger (advice), Mads Sørensen, and Tine Ravn (QRP development).

## Author Contributions

**Conceptualization:** Jesper W. Schneider, Nick Allum, Michael Bang Petersen, Niels Mejlgaard, Robert Zachariae.

**Data curation:** Jesper W. Schneider, Jens Peter Andersen, Emil B. Madsen.

**Formal analysis:** Jesper W. Schneider, Nick Allum, Michael Bang Petersen, Emil B. Madsen.

**Funding acquisition:** Jesper W. Schneider.

**Investigation:** Jesper W. Schneider.

**Methodology:** Jesper W. Schneider, Jens Peter Andersen, Michael Bang Petersen, Niels Mejlgaard, Robert Zachariae.

**Project administration:** Jesper W. Schneider.

**Software:** Jens Peter Andersen.

**Visualization:** Emil B. Madsen.

**Writing – original draft:** Jesper W. Schneider.

**Writing – review & editing:** Jesper W. Schneider, Michael Bang Petersen, Emil B. Madsen, Niels Mejlgaard, Robert Zachariae.

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
