## [Decision Letter · Decision Letter 0]

20 Nov 2023

PONE-D-23-29856Is something rotten in the state of Denmark? Cross-national evidence for widespread involvement but not systematic use of questionable research practices across all fields of researchPLOS ONE

Dear Dr. Schneider,

Thank you for submitting your manuscript to PLOS ONE. After careful consideration, we feel that it has merit but does not fully meet PLOS ONE’s publication criteria as it currently stands. Therefore, we invite you to submit a revised version of the manuscript that addresses the points raised during the review process.

Thank you for submitting your manuscript to PLOS ONE. I apologize for the delay in our review process. I must inform you that your manuscript will require major revisions before we can consider it for publication. The evaluations provided by our first two reviewers were somewhat contradictory; Reviewer 1 recommended "Major Revision," while Reviewer 2 recommended "Accept." However, it is important to note that Reviewer 1 recognizes the high value and potential of your manuscript for publication. Therefore, I would like you to carefully consider and address the concerns raised by Reviewer 1, particularly regarding the methodology of data analysis and its presentation.

We look forward to receiving your revised manuscript.

Kind regards,

Kyoshiro Sasaki, Ph.D.

Academic Editor

PLOS ONE

 [JWS 6183-00001B Danish Agency for Science and Higher Education (Ministry of Higher Education and Science)

https://ufm.dk/forskning-og-innovation/tilskud-til-forskning-og-innovation/hvem-har-modtaget-tilskud/2016/bevilling-til-forskning-i-dansk-forskningsintegritet-fra-styrelsen-for-forskning-og-innovation

No].  

[This work is supported by the PRINT project (Practices, Perceptions, and Patterns of Research Integrity) funded by the Danish Agency for Science and Higher Education (Ministry of Higher Education and Science) under grant No 6183-00001B.

We thank Kaare Aagaard; Asger Dalsgaard Pedersen; Pernille Bak Pedersen (survey development), Emil B. Madsen (technical assistance), Allan Rye Lyngs (web scraping); Ana Marusic; Nicole Foeger (advise); Mads Sørensen, Tine Ravn (QRP development).]

 [JWS 6183-00001B Danish Agency for Science and Higher Education (Ministry of Higher Education and Science)

https://ufm.dk/forskning-og-innovation/tilskud-til-forskning-og-innovation/hvem-har-modtaget-tilskud/2016/bevilling-til-forskning-i-dansk-forskningsintegritet-fra-styrelsen-for-forskning-og-innovation

No].

4. We notice that your supplementary table S1 are included in the manuscript file. Please remove them and upload them with the file type 'Supporting Information'. Please ensure that each Supporting Information file has a legend listed in the manuscript after the references list.

6. Please include a copy of Table S2 in supporting information which you refer to in your text.

Reviewers' comments:

Reviewer's Responses to Questions

**Comments to the Author**

1. Is the manuscript technically sound, and do the data support the conclusions?

Reviewer #1: Partly

Reviewer #2: Yes

2. Has the statistical analysis been performed appropriately and rigorously? 

Reviewer #1: No

Reviewer #2: Yes

3. Have the authors made all data underlying the findings in their manuscript fully available?

Reviewer #1: Yes

Reviewer #2: Yes

4. Is the manuscript presented in an intelligible fashion and written in standard English?

Reviewer #1: Yes

Reviewer #2: Yes

5. Review Comments to the Author

Reviewer #1: The manuscript proposes to examine the use and prevalence of 25 Questionable Research Practices (QRPs) in a large sample of Danish researchers from various academic fields, and compare it to a sample of researchers from 4 other countries. The sample for this study is quite large and the diversity of academic areas, institutions, and locations, as well as the broader list of practices examined, makes for some very interesting descriptive data on QRPs. The authors certainly have what they need here to write a great manuscript that adds to this literature. However, the current manuscript includes unnecessary information and figures, and a few uninterpretable analyses due to invalid aggregation procedures, that obscure these interesting descriptive findings. Below, I've detailed two major issues, several “medium” issues, and a few minor ones that I believe could be addressed to substantially improve this manuscript, and allow the reader better access to the descriptive findings in these data.

Major issues

1. Using “QRP scores” that aggregate different QRP items for each respondent, resulting in invalid analyses and comparisons among groups. The authors have an initial pool of 25 QRP items, all of which, presumably, have different true scores for use and prevalence in the population. However, these items are not all applicable to every researcher surveyed, as many pertain to practices that may not be used in a given field of research (e.g., historians will not p-hack, because p-values are probably not used in their field). To get around this problem, the study separated the respondents into 4 “preferred research approach” groups; according to the supplement, some items could be presented to every researcher (e.g., QRPs related to authorship or reviewing) and others only to a subset of researchers (e.g., QRPs related to significance testing only shown to researchers who reported their field uses significance testing). A total of 9 items were presented to each researcher, some of which appear to be mandatory depending on research approach and others were randomly sampled from a pool of eligible items, resulting in a measurement instrument for each participant that includes 9 of the 25 items. This in itself is not a problem — it actually solves two problems at once, by only presenting items that are more likely to be relevant to each participant, and reducing participant fatigue. But it has the downside of making aggregation of the responses extremely difficult or impossible: statements like “The median number of self-reported QRPs was three, and the average number 2.7 (DK) and 2.5 (INT). (p. 16)” become meaningless, because they are aggregated across different QRP statements. This would be similar to creating 25 math questions with a wide range of difficulties, then assigning different subsets across 4 classes, and testing each student using a semi-random subset of 9 questions. The average number of correct questions would hardly reflect anything meaningful about how good the students are at math, because they all took different tests with varying levels of difficulty. The tests would be different even within each classroom, so averages within or comparisons among these “preferred research approach” groups are invalid as well. Finally, comparisons across “main scientific fields”, depicted in Figure 2, are also problematic for similar reasons. Therefore, I urge the authors to remove most (or all) of the “aggregate results” section (pp. 16-17), as well as figures 1 and 2, from the manuscript, and not to use the aggregated score (or any form of aggregation of the responses to the 9 QRP statements presented to each individual) in any analyses. If the authors wish to make comparisons about the use or prevalence of QRPs with other studies in the literature, my suggestion is to select the items in this study that best match the items in the comparison study, and make careful item-by-item comparisons (e.g., “20% of respondents in this study admitted to having ever p-hacked, compared to 30% in John et al., and 22% in Smith et al.”). In my opinion, the results section should start with the self-reported use of the 25 individual QRPs, or research question 1 in the preregistration. If differences in QRPs for different “main scientific fields” are interesting enough to have their own figure, they could be displayed as separate estimates for each item and group, in a similar way as the results presented in the current figure 7.

2. The models used in the Predictors of Prevalence section use a “kitchen sink” approach, and an aggregated outcome. I need to preface this point by saying that I don’t know enough about Bayesian statistics to fully understand the models used in the “predictors of prevalence” section, or whether they are suitable for these analyses, so my review doesn't cover that. However, I’ll assume these are Bayesian versions of some form of multiple linear regression, or logistic regression. My specific concern is related to using all the predictors to model the outcome, regardless of how they may relate to each other. This approach makes assumptions about how the predictors are (or aren’t) related, and ignores the (probably very complicated) relationships among these variables. The reported correlations on figures 6 and 7 are (or are akin to) partial correlations, or correlations “controlling” for all other predictors. This makes the interpretation of each coefficient very difficult, if not impossible, and can result in inappropriate estimates (e.g., see https://doi.org/10.1177/25152459221095823). This analysis would be clearer (and, I believe, much more informative) if, instead of this model with all the predictors, the authors presented a correlation matrix for all these variables. We could then see the actual correlation between, say, perceived pressure to publish and the use or QRPs without making assumptions about how those two variables are related to the other 8.

Another issue with this analysis is that the outcome seems to be the same kind of aggregated “QRP usage” I mentioned in my previous point, and if it is, it should not be used as the outcome in this model, for the reasons described above. One alternative might be to report the correlations among all 9 predictors and their correlations with all 25 outcomes. That seems unwieldy at best but maybe there’s some way to present that without it becoming overwhelming. Maybe a correlation matrix among the predictors (to help readers understand which ones are correlated), and a figure similar to figs 3-5 showing the correlation of each QRP item with the 9 predictors? Or maybe a 9x25 heatmap (e.g., https://r-graph-gallery.com/heatmap) could be a good way to present these correlations between the 25 QRPs and the 9 predictors in a more visual way. I hope the authors find that at least one of these solutions works well.

Medium issues

3. Too many QRPs, not enough grouping. The fact that a much wider diversity of practices is examined in this paper is a major strength, but it also makes it difficult to understand the results, especially when all 3 of the main figures (currently figs 3-5) display these practices in a different order. Both in the preregistration and in the paper, the authors suggest ways to group these practices — in the prereg, they mention 12 groups, and in the manuscript, they mention 9(?) groups (lines 5-7 of page 12, right above Table 2). Presenting figures 3-5 using (either of) those groupings, and with QRPs in the same order, would go a long way towards helping the reader digest the results, and would allow for comparison among the three figures.

4. QRP statements. The full list of QRP statements (and which groups of respondents got to see what subset of them) is only included in the supplement, despite being central to this study’s claims; at least some of this information should be included in the main manuscript. The authors can also concurrently introduce whatever grouping of these QRP statements they decide to use in addressing point #3 above.

5. Deviations from the preregistered analysis plan are not clearly indicated. The preregistered analysis plan (https://osf.io/75g3d) includes a few dependent variables and analyses that are not in the paper and/or are only briefly mentioned, without explanation for why they were not included. I couldn’t find DV1 (participants’ “trust in research findings in their field of research”) and DV4 (participants’ “beliefs about how damaging such a QRP may be for the trust in research findings or claims in your specific field of research”) in the paper, there’s a list experiment that only gets a very brief mention in the limitations, and the explanation for why the peer review predictor wasn’t included in the paper isn't clear (more on that below). As they say, “preregistration is a plan, not a prison”; to me, it’s fine to make decisions that we couldn’t have foreseen before data collection, but which will make the paper better, especially with such a large dataset, but please make sure to report and explain all the deviations from the initial plan.

6. Introduction and comparison studies. The review of the literature is extensive, but the most important part of it, in my opinion, are the several studies that tried to estimate the use and prevalence of QRPs, and to which the authors frequently compare their results. Most readers will not be able to recall the information for all these studies, and the fact that references are numeric instead of in APA format (which would include at least the first author's last name and year of publication), made it very hard for me to keep the comparison studies even vaguely in my head as I read the results. One solution that would substantially improve these comparisons (and the introduction) is to include a table somewhere that summarizes the main studies used in comparisons. The table could include things like authors’ names, the sample size, other relevant sample characteristics, the QRPs that were studied, how the use and/or prevalence was elicited, and even the main results/estimates. If that's too much information to include in a single table, a full table is maybe more appropriate for the supplement or OSF. But even a list of which studies the text is referring to and basic characteristics of their methods would go a long way in helping the reader make sense of the comparisons made in the results and discussion sections.

7. Preferred research approach. The authors report that the respondents chose “one preferred research approach among four”—non-empirical, empirical quantitative, empirical quantitative using significance testing, and empirical qualitative—but the survey shows that respondents were funneled into these 4 categories by 3 sequential forced-choice questions: empirical vs. non-empirical approach, quantitative vs. qualitative methods, and a final question, “Are statistical significance tests used in your field of research?”, which is not about whether the researcher themselves employs a quantitative approach using statistical significance tests. “Empirical quantitative” and “empirical quantitative using significance testing” are nested categories and don’t quite reflect the questions respondents were asked, making it more difficult to interpret the findings. I would suggest describing the survey questions in a bit more detail and relabeling the categories accordingly, at least the first time the category is mentioned (e.g., “a researcher that uses primarily empirical, quantitative approaches in a field that uses/doesn’t use significance testing”, which would then not be overlapping categories). Since these categories are, then, not exclusively about the researchers’ methods nor about their fields’ methods, it’s important to describe the distribution of these 4 categories into the researchers’ fields. This will probably be more practical as a figure than a table. (I would suggest that the authors consider making tables 1 and 2 into bar charts as well so the numbers in those tables are more easily comparable.)

8. Ad hoc predictor scales. Using ad hoc scales is not in itself a problem, but it would help if the authors presented a little bit more information on these measures in the main manuscript, especially since they make a decision to omit the preregistered peer review measure entirely (but see below). For example, in the Survey Instrument section, when describing the scales, the authors should include a little more explanation of the constructs they are trying to capture (e.g., what does “local research culture” mean?), an example item from each scale, and some information about the reliability and validity of each measure (e.g., alpha or omega coefficients, maybe fit information for a 1-factor CFA model?).

9. The missing peer review predictor. Without more information about each measure, it’s hard to know why exactly the authors chose to omit the peer review predictor, but they do mention running an exploratory factor analysis and finding that “only two constructs were valid as scales”. I found this confusing; these are all pretty different constructs, and I’m not sure what putting them all together in an EFA would be able to tell us. I’d much rather get some reliability and fit information for a 1-factor CFA on each ad hoc scale, as mentioned above. If that shows that the peer review items have terrible reliability, is one of the items mainly responsible for the problems? The authors could just drop it, explain what happened, and use the variable as preregistered.

Minor issues

10. Were PhD students excluded? It sounds like having completed a PhD was a prerequisite to participating in this study, which is fine, but it would be good to see some more explanation of why this decision was made, and maybe mention this somewhere in the discussion. There may be relevant differences between researchers who have a PhD and those still in training in terms of QRPs and the other variables measured.

11. Add a bit more guidance on the Bayesian stats. It would help readers who are not as familiar with Bayesian stats if the manuscript includes a bit more hand-holding when referring to these models. For example, it says “Jeffrey’s prior, (, = 0.5,0.5)” was used, but why was that the case? What does this prior imply or why is it more adequate than other priors? I don’t mean that Bayesian researchers should always include a “primer” with their manuscripts just because the majority of researchers insists on being frequentists, but some extra bit of explanation for the uninformed like me is always greatly appreciated :)

12. The way QRP prevalence is calculated was a bit confusing. I had to go back a few times and I’m not totally sure I understand it now. Maybe an example would help?

13. More headings please. My final point is more of a stylistic suggestion, but I found it very hard to follow the flow of the paper without more subheadings, especially in the results section. Maybe it would help to move some of the information there to the discussion as well, but I just found it hard to follow the results without more headings, especially for a study with so many variables and multiple whole-page figures. I kept losing my place in the manuscript while I looked at the figures, and a few extra subheadings would probably have helped.

Finally, I would like to congratulate the authors on the time and energy they undoubtedly dispensed in obtaining such an interesting dataset on so many researchers from different countries, and reiterate that I hope the authors will choose to rewrite the manuscript to address the aggregation problem and other outlined issues that detract from the overall findings.

I hope this feedback is helpful, and I’m always happy to clarify if anything in this review is unclear!

Best,

—Julia Bottesini

Reviewer #2: In this manuscript entitled “is something rotten in the state of Denmark? Cross-national evidence for widespread involvement but not systematic use of questionable research practices across all fields of research,” the authors attempt to reveal perception, use, and prevalence of QRPs among researchers across academic fields in Denmark with international comparison. In addition, the predictive factors that lead to QRPs, such as perceived pressure and personality traits, are also examined.

I offer my assessment of this manuscript from the perspective of a Japanese psychologist working on open science. In this manuscript, previous studies are well-reviewed. The research questions are clear, and the approach to them is appropriate. Furthermore, the analysis and the results are clearly shown, and their interpretation is described logically and well supported by the extensive and rigorous datasets. This article is very informative and contributes to the understanding of the reality of QRPs. Therefore, I do not have any major changes to address except for a few minor points. I recommend the publication of this manuscript.

Minor points

1. I believe that including two or three examples for each of the four main scholarly fields mentioned on lines 14 to 16 of page 11 would help readers to understand better.

2. I may be overlooking something, but discussing the generalizability of the study's findings in the discussion or limitations section might be beneficial. In other words, do the findings of this study apply to countries and regions other than Denmark, the UK, the USA, Croatia, and Austria? For instance, the study's participants are primarily researchers from Western countries, but do the authors think similar results would be obtained in Asian countries like Japan? This is merely a personal interest of mine and may be slightly out of the scope of the study, so it is not necessary to address it.

6. PLOS authors have the option to publish the peer review history of their article (what does this mean?). If published, this will include your full peer review and any attached files.

Reviewer #1: **Yes: **Julia G. Bottesini

Reviewer #2: No

---

## [Author Response · Author response to Decision Letter 0]

28 Feb 2024

Response to Reviewers

[1] When submitting your revision, we need you to address these additional requirements.

Response:

We have tried to ensure that the manuscript and files follow PLOS ONE’S style requirements.

[2] Thank you for stating the following financial disclosure: 

[JWS 6183-00001B Danish Agency for Science and Higher Education (Ministry of Higher Education and Science) https://ufm.dk/forskning-og-innovation/tilskud-til-forskning-og-innovation/hvem-har-modtaget-tilskud/2016/bevilling-til-forskning-i-dansk-forskningsintegritet-fra-styrelsen-for-forskning-og-innovation No]. 

Please state what role the funders took in the study. If the funders had no role, please state: “The funders had no role in study design, data collection and analysis, decision to publish, or preparation of the manuscript.”

Response:

We have added “The funders had no role in study design, data collection and analysis, decision to publish, or preparation of the manuscript”.

We have included this amended Role of Funder statement in our revised cover letter.

[3] Thank you for stating the following in the Acknowledgments Section of your manuscript: 

[This work is supported by the PRINT project (Practices, Perceptions, and Patterns of Research Integrity) funded by the Danish Agency for Science and Higher Education (Ministry of Higher Education and Science) under grant No 6183-00001B.

We thank Kaare Aagaard; Asger Dalsgaard Pedersen; Pernille Bak Pedersen (survey development), Emil B. Madsen (technical assistance), Allan Rye Lyngs (web scraping); Ana Marusic; Nicole Foeger (advise); Mads Sørensen, Tine Ravn (QRP development).]

[JWS 6183-00001B Danish Agency for Science and Higher Education (Ministry of Higher Education and Science)https://ufm.dk/forskning-og-innovation/tilskud-til-forskning-og-innovation/hvem-har-modtaget-tilskud/2016/bevilling-til-forskning-i-dansk-forskningsintegritet-fra-styrelsen-for-forskning-og-innovation No].

Response:

We have removed all funding-related text from the manuscript.

We would like the Fundings Statement to be the following:

[This work is supported by the PRINT project (Practices, Perceptions, and Patterns of Research Integrity) funded by the Danish Agency for Science and Higher Education (Ministry of Higher Education and Science) under grant No JWS 6183-00001B]

The Acknowledgment Section is changed to:

[We thank Kaare Aagaard; Asger Dalsgaard Pedersen; Pernille Bak Pedersen (survey development), Allan Rye Lyngs (web scraping); Ana Marusic; Nicole Foeger (advise); Mads Sørensen, Tine Ravn (QRP development).]

Notice, we have removed “Emil B. Madsen (technical assistance)” from the acknowledgements and provided him with an authorship instead due to his strong involvement in revising the manuscript.

We have included the amended funder statement and change of authorship in our revised cover letter.

[4] We notice that your supplementary table S1 are included in the manuscript file. 

Please remove them and upload them with the file type 'Supporting Information'. Please ensure that each Supporting Information file has a legend listed in the manuscript after the references list.

Response:

We have now upgraded Table S1 to the manuscript.

[5] Please include captions for your Supporting Information files at the end of your manuscript, and update any in-text citations to match accordingly. 

Please see our Supporting Information guidelines for more information: http://journals.plos.org/plosone/s/supporting-information. 

Response:

This has been done.

[6] Please include a copy of Table S2 in supporting information which you refer to in your text.

Response:

This was a mistake. We have updated all supporting information according to the revisions suggested by Reviewer 1 and provide a new set of files listed in the Supporting Information.

 

Responses to Reviewer 1

We would like to thank Reviewer 1 for relevant and incisive comments. We have attempted to address them to the best of our ability below.

[1] Major: Using “QRP scores” that aggregate different QRP items for each respondent, resulting in invalid analyses and comparisons among groups. The authors have an initial pool of 25 QRP items, all of which, presumably, have different true scores for use and prevalence in the population. However, these items are not all applicable to every researcher surveyed, as many pertain to practices that may not be used in a given field of research (e.g., historians will not p-hack, because p-values are probably not used in their field). To get around this problem, the study separated the respondents into 4 “preferred research approach” groups; according to the supplement, some items could be presented to every researcher (e.g., QRPs related to authorship or reviewing) and others only to a subset of researchers (e.g., QRPs related to significance testing only shown to researchers who reported their field uses significance testing). A total of 9 items were presented to each researcher, some of which appear to be mandatory depending on research approach and others were randomly sampled from a pool of eligible items, resulting in a measurement instrument for each participant that includes 9 of the 25 items. This in itself is not a problem — it actually solves two problems at once, by only presenting items that are more likely to be relevant to each participant, and reducing participant fatigue. But it has the downside of making aggregation of the responses extremely difficult or impossible: statements like “The median number of self-reported QRPs was three, and the average number 2.7 (DK) and 2.5 (INT). (p. 16)” become meaningless, because they are aggregated across different QRP statements. This would be similar to creating 25 math questions with a wide range of difficulties, then assigning different subsets across 4 classes, and testing each student using a semi-random subset of 9 questions. The average number of correct questions would hardly reflect anything meaningful about how good the students are at math, because they all took different tests with varying levels of difficulty. The tests would be different even within each classroom, so averages within or comparisons among these “preferred research approach” groups are invalid as well. Finally, comparisons across “main scientific fields”, depicted in Figure 2, are also problematic for similar reasons. Therefore, I urge the authors to remove most (or all) of the “aggregate results” section (pp. 16-17), as well as figures 1 and 2, from the manuscript, and not to use the aggregated score (or any form of aggregation of the responses to the 9 QRP statements presented to each individual) in any analyses. If the authors wish to make comparisons about the use or prevalence of QRPs with other studies in the literature, my suggestion is to select the items in this study that best match the items in the comparison study, and make careful item-by-item comparisons (e.g., “20% of respondents in this study admitted to having ever p-hacked, compared to 30% in John et al., and 22% in Smith et al.”). In my opinion, the results section should start with the self-reported use of the 25 individual QRPs, or research question 1 in the preregistration. If differences in QRPs for different “main scientific fields” are interesting enough to have their own figure, they could be displayed as separate estimates for each item and group, in a similar way as the results presented in the current figure 7.

Response:

We thank Reviewer 1 for this pivotal methodological criticism. We agree that aggregating the 25 different QRPs in the way we have done may be problematic. 

In the revised manuscript we follow Reviewer 1’s suggestion and completely remove the “aggregate results” section including figures 1 and 2, as well as the aggregate results relating to “main scientific fields”.

For the regression models predicting self-reported prevalence, we now base all models on analyses among respondents within the same “preferred research approaches” (see also comment 2b below). Here, Reviewer 1 also expresses some skepticism, but we do believe this is a much stronger design. Within the “preferred research approaches”, respondents here have been subjected to a much more consistent common pool of QRPs. This commonality allows for meaningful comparisons and analyses within each group, as the basis for comparison is uniform. Some statements are mandatory and others are randomly assigned within the constraints of each preferred approach, but – and this is key – within each preferred approach the random items are drawn from the same pool of items for all participants. Given that randomization is effective, then the effects of random assignment and missing items will cancel out at the aggregate level of each approach (this is essentially the logic of any missing-by-design questionnaire designs). This allows for a valid comparison of response patterns within each preferred approach group. By analyzing responses within each approach, we are inherently controlling for the variation in QRP exposure. This makes it easier to interpret differences or similarities in response patterns as being more directly related to the respondents’ perceptions and behaviors rather than the variability in which QRPs they were asked about.

On this basis, the revised manuscript analyses data at the level of preferred research approaches as it accounts for the structured variability in QRP exposure. We therefore use this approach for modelling the predictors. At the same time, to accommodate the concerns of Reviewer 1 we reproduce our analyses only focusing on the set of 11 QRP items eligible to all participants. Reassuringly, this analysis replicates the main findings from analyses broken down into the four preferred research approaches.

[2a] Major: The models used in the Predictors of Prevalence section use a “kitchen sink” approach, and an aggregated outcome. 

I need to preface this point by saying that I don’t know enough about Bayesian statistics to fully understand the models used in the “predictors of prevalence” section, or whether they are suitable for these analyses, so my review doesn't cover that. However, I’ll assume these are Bayesian versions of some form of multiple linear regression, or logistic regression. My specific concern is related to using all the predictors to model the outcome, regardless of how they may relate to each other. This approach makes assumptions about how the predictors are (or aren’t) related and ignores the (probably very complicated) relationships among these variables. The reported correlations on figures 6 and 7 are (or are akin to) partial correlations, or correlations “controlling” for all other predictors. This makes the interpretation of each coefficient very difficult, if not impossible, and can result in inappropriate estimates (e.g., see https://doi.org/10.1177/25152459221095823). This analysis would be clearer (and, I believe, much more informative) if, instead of this model with all the predictors, the authors presented a correlation matrix for all these variables. We could then see the actual correlation between, say, perceived pressure to publish and the use or QRPs without making assumptions about how those two variables are related to the other.

Response:

We thank Reviewer1 for this observation. 

First, the Linear Probability Model (LPM) used in the manuscript is essentially a linear regression applied to bounded outcome data between 0 and 1. We have checked to what extent this model is useful by also applying fractional regression with a logit model to the data. The results are similar in which case the LPM seems more interpretable as it provides results on the outcome scale, whereas fractional regression reports log-odds which are harder to interpret. 

Second, we do not think our model can be designated as a “kitchen sink model”. The latter is typically used to imply that a long list of possible independent variables (IVs) is thrown into a model without much consideration for theoretical justification, relevance, risk of multicollinearity etc., in hope of finding some statistical pattern usually defined as IVs being statistically significant.

As we argue in the manuscript, our models are exploratory, and we are not postulating any causal relations or make claims based on “statistical significance”. Nevertheless, it is certainly true that the interpretation of the model coefficients depends on the potential causal relationship between them, and that a full model with all predictors makes interpretation of each coefficient prone to “Table 2 fallacy”, where each is interpreted as a direct effect, even if the causal structure of the model lends itself to a myriad of indirect relationships. However, we do not think this is the case here. We specifically control for gender and academic age (years after PhD) to avoid their possible confounding effects, as engagement in QRPs, the OCEAN factors, and perceived pressure are known to vary according to both age and gender. In the revised figures 5-7, we also present the estimates for each predictor from a simple bivariate model (incl. only that predictor) for comparison with the estimates from the full model.

We also argue that when the selection of variables is deliberate, and there’s a clear interest in examining the full model’s partial coefficients without engaging in a fishing expedition for statistical significance, this approach can be viewed as thorough rather than indiscriminate.

Obviously, it can be discussed to what extent there is a theoretical justification for this, but as we argue this is a first attempt to bring these different sets of predictors together, not some kind of confirmatory causal analysis, and as such all covariates considered are plausible and chosen based on substantive grounds (i.e. previous research) for exploration in a descriptive and exploratory manner. We therefore think that our model is justified based on previous findings and suggestions.

We agree with Reviewer 1 that our choice of predictors should be more clearly motivated. We have therefore revised this in the Analysis subsection on page 20 [manus with track-changes]:

““All predictors were carefully chosen. As already indicated, the systemic factors included have high predictive relevance since pressures are often cited as a cause for breaches in integrity [e.g., 30] and a supportive research culture is often pointed to as a plausible protective [e.g., 59]. Likewise, gender and academic age have also been examined before although their predictive value is unclear [e.g., 76]. What is novel in our approach is to combine these relevant systemic and demographic predictors with personality factors in the normal range. We conjecture that propensities for engagements in QRPs are a complicated mixture of experience, systemic and individual factors.”

We also acknowledge that we should provide more information in relation to the limitations and challenges associated with including many predictors. As suggested by Reviewer 1, we have therefore provided a correlation matrix between all predictor variables, provided information about potential multicollinearity (none) and as stated above, also provided bivariate coefficients in the new plots in order to compare changes between a bivariate coefficient and its partial equivalent, i.e. when conditioning on the other c

---

## [Decision Letter · Decision Letter 1]

3 Apr 2024

PONE-D-23-29856R1Is something rotten in the state of Denmark? Cross-national evidence for widespread involvement but not systematic use of questionable research practices across all fields of researchPLOS ONE

Dear Dr. Schneider,

Thank you for submitting your manuscript to PLOS ONE. After careful consideration, we feel that it has merit but does not fully meet PLOS ONE’s publication criteria as it currently stands. Therefore, we invite you to submit a revised version of the manuscript that addresses the points raised during the review process.

We look forward to receiving your revised manuscript.

Kind regards,

Kyoshiro Sasaki, Ph.D.

Academic Editor

PLOS ONE

Journal Requirements:

Additional Editor Comments:

Thank you for submitting your revised manuscript. We appreciate the time and effort you've invested in addressing the concerns previously outlined by our reviewers. Upon re-evaluation, the reviewer who assessed the initial submission has expressed satisfaction with your revisions and has suggested a few minor points for consideration. I concur with the reviewer that these points merit attention.

Therefore, my decision is "Minor Revision" – meaning, your manuscript is accepted subject to the proposed revisions being addressed.

Reviewers' comments:

Reviewer's Responses to Questions

**Comments to the Author**

1. If the authors have adequately addressed your comments raised in a previous round of review and you feel that this manuscript is now acceptable for publication, you may indicate that here to bypass the “Comments to the Author” section, enter your conflict of interest statement in the “Confidential to Editor” section, and submit your "Accept" recommendation.

Reviewer #1: (No Response)

2. Is the manuscript technically sound, and do the data support the conclusions?

Reviewer #1: Yes

3. Has the statistical analysis been performed appropriately and rigorously? 

Reviewer #1: Yes

4. Have the authors made all data underlying the findings in their manuscript fully available?

Reviewer #1: Yes

5. Is the manuscript presented in an intelligible fashion and written in standard English?

Reviewer #1: Yes

6. Review Comments to the Author

**Reviewer #1:** The revised manuscript offers a much clearer, transparent, and informative description of the study and its results, highlighting the important descriptive work that was done. I commend the authors for being willing to make these revisions and taking my suggestions seriously. It’s rewarding when the peer review process results in what I feel is a better version of the manuscript — and I hope the authors feel similarly.

In particular, Figure 1 combined with Table 1 are excellent additions and really help explain the method used in the study. Although I still disagree with the authors’ decision not to present the QRPs in the same order (and/or grouped by category) in Figures 2-4, I feel like they have a perfectly valid reason for doing so. I was also initially skeptical that the results could be aggregated in a valid way within “research approach” groups, but was convinced by the authors’ explanation of this. The results and discussion sections never seem to make any comparisons among these groups, which is reassuring.

I was also pleased with the changes which relate to the preregistration. The authors have now included a clearer explanation of which analyses were preregistered and what the deviations from the preregistration are, which makes the study more transparent. Furthermore, although the preregistration remains available and linked in the manuscript (as it should), the study itself is not described as preregistered in the abstract. I want to point out that I believe this is the right move here; the extensive deviations from the preregistration, and the fact that the preregistration applies to a larger dataset than described, leave this study in a bit of a gray area. Including the preregistration but not describing the study as preregistered feels like the most appropriate solution.

Overall, the authors have addressed most of my concerns, whether by making changes to the manuscript or providing a convincing explanation for why they decided to do something else. Any lingering issues are matters of opinion or personal preference, and therefore it is clearly the prerogative of the authors to decide how they should be dealt with.

I have two final suggestions for the discussion section, which I am happy for the authors to incorporate or ignore. First, I still think the citations as numbers makes it a lot harder to understand and remember which study the authors are referring to. The table that was included in the supplement helps with this, but it’s not easily accessible. Instead, I would suggest slightly rephrasing the sentences to include the name of the authors, at least in the discussion if not the whole manuscript. Here’s an example sentence from page 33, lines 10-11, with edits in brackets: “They are also in line with the pooled estimate [obtained by Fanelli] [62], but much higher than the recent estimate [by Xie et al.] [61].”

My second suggestion is to edit the last couple of paragraphs in the paper for a tighter, more effective conclusion — perhaps even adding one last subheading (“Conclusion”) right before the Mazar paragraph? It’s fine as it is and I like the reference to Mazar and colleagues’ claim, it just needs a bit of reworking to be clearer as it’s currently a little difficult to follow. Relatedly, the manuscript has a few typos here and there that the authors might want to address by pasting the final manuscript text into a google doc or some other text-checking tool — it’s certainly nothing that compromises comprehension, though. I only mention this because the PLoS review system says they do not copyedit.

Once again, I’d like to thank the authors for a good peer-reviewing experience, which is rarer than it should be. I hope the process was useful for them as well, and I look forward to reading the published paper!

If anything in this review is unclear, please feel free to reach out to me directly.

Best,

—Julia Bottesini

7. PLOS authors have the option to publish the peer review history of their article (what does this mean?). If published, this will include your full peer review and any attached files.

Reviewer #1: **Yes: **Julia G. Bottesini

---

## [Author Response · Author response to Decision Letter 1]

5 May 2024

Response to reviewers

We want to thank Reviewer 1 and the Editor for further relevant comments. We have attempted to address them to the best of our ability below.

Editor comment:

• Please review your reference list to ensure that it is complete and correct. Any changes to the reference list should be mentioned in the rebuttal letter that accompanies your revised manuscript.

Response: We have checked the reference list and identified four duplicates (see below), which means that the reference list has been reduced from 113 to 109. This also means that the numbering has changed. We have updated that in the manuscript and in the supplementary files.

Duplicate references Old numbers Revised new number

Edwards #29; #30 #29

Makel #68; #71 #68

Bakker #70; #74 #69

Chien #72; #95 #70

Reviewer 1 comments:

I have two final suggestions for the discussion section, which I am happy for the authors to incorporate or ignore. 

• First comment: First, I still think the citations as numbers makes it a lot harder to understand and remember which study the authors are referring to. The table that was included in the supplement helps with this, but it’s not easily accessible. Instead, I would suggest slightly rephrasing the sentences to include the name of the authors, at least in the discussion if not the whole manuscript. Here’s an example sentence from page 33, lines 10-11, with edits in brackets: “They are also in line with the pooled estimate [obtained by Fanelli] [62], but much higher than the recent estimate [by Xie et al.] [61].”

Response: From the results section onwards we indicated the reference compared to in the manner suggested by Reviewer 1.

• Second comment: My second suggestion is to edit the last couple of paragraphs in the paper for a tighter, more effective conclusion — perhaps even adding one last subheading (“Conclusion”) right before the Mazar paragraph? It’s fine as it is and I like the reference to Mazar and colleagues’ claim, it just needs a bit of reworking to be clearer as it’s currently a little difficult to follow. 

Response: We have rewritten the last paragraphs so that our argument (hopefully) comes through more clearly, and we have also created a ‘Conclusion’ subheading, as suggested.

• Third comment: Relatedly, the manuscript has a few typos here and there that the authors might want to address by pasting the final manuscript text into a google doc or some other text-checking tool — it’s certainly nothing that compromises comprehension, though. I only mention this because the PLoS review system says they do not copyedit.

Response: We have scrutinized the manuscript for typos and commas and had native English speakers check the grammar. Hopefully, we have identified the most errors and thus made the manuscript more readable.

Again, many thanks for the comments.

---

## [Editor Report · Decision Letter 2]

10 May 2024

Is something rotten in the state of Denmark? Cross-national evidence for widespread involvement but not systematic use of questionable research practices across all fields of research

PONE-D-23-29856R2

Dear Dr. Schneider,

We’re pleased to inform you that your manuscript has been judged scientifically suitable for publication and will be formally accepted for publication once it meets all outstanding technical requirements.

Kind regards,

Kyoshiro Sasaki, Ph.D.

Academic Editor

PLOS ONE

Additional Editor Comments (optional):

Thank you for revising the manuscript. Your manuscript has been improved and is suitable for publication in PLOS ONE.
---

## [Editor Report · Acceptance letter]

21 Jun 2024

PONE-D-23-29856R2 

PLOS ONE

Dear Dr. Schneider, 

I'm pleased to inform you that your manuscript has been deemed suitable for publication in PLOS ONE. Congratulations! Your manuscript is now being handed over to our production team.

Kind regards, 

on behalf of

Dr. Kyoshiro Sasaki 

Academic Editor

PLOS ONE